



# Ozone source attribution in polluted European areas during summer as simulated with MECO(n)

Markus Kilian[1], Volker Grewe[1,3], Patrick Jöckel[1], Astrid Kerkweg[2], Mariano Mertens[1], Andreas Zahn[4], and Helmut Ziereis[1]

[1]Deutsches Zentrum für Luft- und Raumfahrt (DLR), Institut für Physik der Atmosphäre, Oberpfaffenhofen, Germany
[2]Institute of Energy and Climate Research, IEK-8: Troposphere, Forschungszentrum Jülich, Jülich, Germany
[3]Delft University of Technology, Faculty of Aerospace Engineering, Section Aircraft Noise and Climate Effects, Delft, The Netherlands
[4]Institute of Meteorology and Climate Research, Forschungszentrum Karlsruhe, Germany

**Correspondence:** Markus Kilian (markus.kilian@dlr.de)

**Abstract.** Emissions of land transport and anthropogenic non-traffic emissions (e.g. industry, households and power generation) are significant sources of nitrogen oxides, carbon monoxide, and volatile organic compounds. These emissions are important precursors of tropospheric ozone and affect air quality. The contribution of emission sectors to ozone cannot be measured directly, but calculated with sophisticated models of atmospheric chemistry only. For this study we apply a the MECO(n)
model system (MESSy-fied ECHAM and COSMO models nested n times) equipped with a source attribution method to investigate the contribution of anthropogenic (land transport and non-traffic) and biogenic emissions to ozone in Europe. This model system couples a global chemistry-climate mode with a regional chemistry-climate model. Our source attribution (tagging) method fully decomposes the budgets of ozone and ozone precursors into contributions from various emission sources and regions. To estimate also the contributions of regional versus long-range transported contributions we distinguish four dif-
ferent source regions: Europe, North America, East Asia and Rest of the Worl. We performed one simulation covering 2 years with two regional refinements, one covering Europe (50 km resolution), and one covering Central Europe (12 km resolution). The model results are evaluated with data from European air quality stations and in situ data from the flight campaign Effect of Megacities on the Transport and Transformation of Pollutants on the Regional to Global Scales (EMeRGe) Europe in Summer 2017. Two study areas with large anthropogenic emissions, Benelux and Po Valley, are compared in detail. The absolute contri-
butions of European land transport emissions to ground-level ozone for JJA 2017 in the Po Valley are larger than in the Benelux region (7 nmol mol$^{-1}$ and $\approx$ 3 nmol mol$^{-1}$), the same applies for the relative contributions with 12 % in the Po Valley and 7 % in the Benelux regions. Similar results are found for the contribution of European anthropogenic non-traffic emissions. Here, absolute contributions are larger in the Po Valley with 11 nmol mol$^{-1}$ (19 %) than 5 nmol mol$^{-1}$ (15 %) in the Benelux regions. The relative contributions to ozone from long-range transported land transport emissions in both regions in the range
of 5–6 %, and the relative contributions from long-range transported non-traffic emission sare 9 % in the Po Valley and 13 % in the Benelux region. Contributions to ozone from long-range transported emissions are clearly more homogeneously distributed throughout Europe, whereas the distribution of contributions to ozone from European emissions is notably in-homogeneous. During periods of high ozone, contributions of European land transport and anthropogenic non-traffic emissions increase in





particular over the Po Valley and in the Benelux. Especially in the Po Valley the increase is very strong and extreme ozone values could be mitigated in the Po Valley by reducing anthropogenic emissions.

# 1 Introduction

Tropospheric ozone contributes to global warming by absorption of radiation (Myhre et al., 2013) and it is harmful for human
health and plants (World Health Organisation, 2003; Jimenez-Montenegro et al., 2021). Extreme ozone events mostly occur during heat waves in major polluted areas and have detrimental effects on human health. The most important sources of tropospheric ozone are the downward transport from the stratosphere and the in situ production from precursors such as carbon monoxide (CO), methane ($CH_4$), nitrogen oxides ($NO_x = NO_2 + NO$), and volatile organic compounds (VOCs; Haagen-Smith (1952); Monks (2005)). VOCs comprise a wide range of compounds, which contain carbon and hydrogen atoms, where the non-
methane compounds are summarised as non-methane hydrocarbons (NMHCs). The ozone precursors arise from anthropogenic sources, such as land transport (railway, inland navigation, and road traffic, Mertens et al., 2020b; Hoor et al., 2009), industry (Ou et al., 2020), and shipping (e.g. Jonson et al., 2020; Matthias et al., 2016; Aulinger et al., 2016; Eyring et al., 2010) as well as from natural sources, such as lightning (Hauglustaine et al., 2001; Schumann and Huntrieser, 2007), wildfires (Di Carlo et al., 2015), and soil bacteria (Yienger and Levy II, 1995; Vinken et al., 2014). In particular the natural sources are subject
to large uncertainties, because emissions can not be directly measured and processes determining the emission fluxes are not yet fully understood (Tost et al., 2007; Mebust et al., 2011; Yienger and Levy II, 1995; Vinken et al., 2014). Since the ozone chemistry is nonlinear (Seinfeld and Pandis, 2006), the contribution of different precursors to ozone can only be estimated with the help of numerical models.

In order to estimate the influence of different emission sectors on tropospheric ozone, either the perturbation method or
the source attribution method can be applied. The perturbation method investigates the change of ozone due to an emission reduction (or increase) by comparing the results of a reference simulation with those of a simulation with changed emissions. This yields the impact of the reduced/increased emission sector on ozone. The source attribution method decomposes ozone and ozone precursors into the shares from various emission sources. This share is the contribution of the emission sector. Due to their different concepts, both methods answer different questions. Table 1 of Mertens et al. (2020b) clarifies, which scientific
questions can be answered by impacts (using a perturbation method) and contributions (calculated by a source attribution method such as tagging). Grewe et al. (2010), Clappier et al. (2017), and Wang et al. (2009) compared the tagging method with the perturbation method. They found, that the perturbation method is inappropriate for source attribution, because it quantifies the change of $O_3$ due to an emission change. In contrast, the source attribution method does not provide information about the sensitivity of $O_3$ to an emission change. For this reason, the perturbation method is rather appropriate to address future
emission policies, than calculating the sector-wise contributions to tropospheric $O_3$. Thus, for studies which aim to distinguish between different emission sources and the spatial origin of the emissions, the source attribution method is indispensable.

Many studies used the perturbation method to quantify the impact of land transport emissions on $O_3$ (see also overview by Mertens et al., 2018). Granier and Brasseur (2003) applied a 100 % perturbation for road transport on the global scale and



found out that during the summer months, the impact on ground-level $O_3$ of road traffic emissions is almost 15 % in Europe. Matthes et al. (2007) stated that in July 1990 road traffic emissions from $NO_x$, CO, and NMHCs have an impact of up to 16 % to ground-level $O_3$ in polluted regions such as central Europe. Similarly, Hoor et al. (2009) reported an impact of road transport on $O_3$ of up to 16 % in the northern hemisphere boundary layer during summer. They applied a 5 % perturbation and scaled
the results to 100 %.

In order to estimate contributions to ground-level $O_3$ by different sectors, other studies applied the source attribution method, which uses tagged tracers for ozone and ozone precursors. Dahlmann et al. (2011) applied a source attribution method to calculate the ozone contribution for individual $NO_x$ emission sources from different sectors. They report that, due to their large $NO_x$ emissions, anthropogenic sources such as road traffic, shipping, and industry largely contribute to the tropospheric ozone
column in the northern mid latitudes. Mertens et al. (2020b) investigated the contribution of emissions from land transport, anthropogenic non-traffic and biogenic sources to ground-level ozone in Europe. They found biogenic emissions, and anthropogenic non-traffic emissions to be the two most important contributors to ground-level $O_3$ in Europe with 19 % and 16 %, respectively. Especially in the Po Valley they report large contributions form land transport emissions. Further, they showed that during events with large ozone mixing ratios the contribution of land transport peaks up to 28 %. Similarly, Lupaşcu and
Butler (2019) quantified the contribution of different emission sectors and regions on ozone in Europe using a $NO_x$ tagging approach. They investigated contributions for different exceedances metrics. They found that local emission sources account for 41 % and 38 % to MDA8 ozone exceedance days in the Po Valley and Germany, respectively.

Altogether, there are many studies applying source attribution methods in global and regional chemistry-climate models. However, based on the current state of research, there is a lack of studies separating ozone contributions into shares from
long-range transported and regional emissions. Therefore, the goal of our study is to close this gap of knowledge with a focus on Europe. Thereby, we consider those emission sectors with the largest ground-level ozone share, because mitigating the emissions of these sectors have the largest potential in mitigating ozone. Hereby, we investigate the two major polluted regions Po Valley and Benelux in detail. We choose these two regions as their meteorological conditions (temperature, wind, sunshine etc.) and amount of biogenic emissions differs. The Benelux region is characterized by a maritime climate dominated by
westerly winds, which favours long-range transport of emissions from Northern America. In contrast, the Po Valley is a basin with a humid continental to humid subtropical climate. The hot dry summers together with large industry and land transport emissions can be considered as perfect conditions for ozone formation.

In order to asses the contributions of different emission sources to ground-level $O_3$ in various regions of Europe, the source attribution method is applied by using the MECO(n) (MESSy-fied ECHAM and COSMO models nested n times) model.
MECO(n) is an on-line coupled global/regional chemistry-climate model, which allows a regionally finer resolutions in order to understand regional processes better. The global model is important, to represent the long-range transport across the boundaries of the embedded regional model. To attribute anthropogenic emissions either as regional or long-range transported, four major source regions (Europe, North America, East-Asia, and rest of the World) are defined. All of these regions have large anthropogenic emissions, which also impact other geographical regions.



We apply the tagging method described by Grewe et al. (2017) and Rieger et al. (2018), which allows us to quantify the contributions of reactive nitrogen ($NO_y$, see Supplement Sect. 2 for a definition), CO and VOC emissions to $O_3$. In this study the major focus lies on $NO_y$ and $O_3$. To evaluate the simulated mixing ratios of $O_3$, $NO_x$ and $NO_y$ we used various data-sets. First, a data fusion product by Delang et al. (2021), based on the observations of the TOAR (Tropospheric Ozone Assessment

Report, https://igacproject.org/activities/TOAR) database and other model results ($O_3$ ground-level measurements from air quality stations from the AirBase (European air quality database) network ($NO_x$ and $O_3$) and measurements during the "Effect of Megacities on the Transport and Transformation of Pollutants at Regional and Global Scales" (EMeRGe) Europe campaign (Andrés Hernández et al., 2022) with the High Altitude and LOng Range Research Aircraft (HALO, $O_3$ and $NO_y$).

The aim of our analyses is to answer the following questions:

– How do the various emission sectors contribute to $NO_y$ and $O_3$ in the Po Valley, and how does this differ in comparison to the Benelux region?

–   How large are the contributions from European emissions compared to the contributions from long-range transported emissions to ground-level $O_3$?

This manuscript is structured as follows: Section 2 provides an overview of the model system and explains the model setup

applied for the simulations. In Section 3 we present an evaluation of the MECO(n) data with other model data, air quality station measurements, and measurements from the HALO research aircraft. The source attribution results are presented in Section 4. Finally, the results are discussed in detail in Section 5 and Section 6 summarizes the most important results and answers the research questions.

## 2   Model Simulations

## 2.1   Model Description

For the present study we apply the MECO(n) model system, which couples two model components on-line: the global chemistry-climate model EMAC (Jöckel et al., 2010, 2016) and the regional chemistry-climate model COSMO-CLM/MESSy (version COSMO 5.0.0 clm16; Kerkweg and Jöckel (2012). The core atmospheric model used in COSMO-CLM/MESSy is the COSMO-CLM model (Rockel et al., 2008), a regional atmospheric climate model that is based on the COSMO (Consortium

for Small-scale Modelling) model and jointly further developed by the CLM-Community. We used EMAC (ECHAM5 version 5.3.02) in the T42L90MA resolution, i.e. with a spherical truncation of T42 (corresponding to a quadratic Gaussian grid of ca. 2.8° x 2.8° in latitude and longitude) with 90 vertical hybrid pressure levels up to 0.01 hPa. EMAC is operated with a timestep length of 720s. For the simulations we used MESSy in the version 2.55.2-1913. Comparable with the study by Mertens et al. (2020a), a MECO(2) set-up with one COSMO/MESSy instance over Europe with a resolution of 0.44° x 0.44° (≈ 50 km,

named CM50), and a further nested instance covering Central Europe with a resolution of 0.11° x 0.11° (≈ 12 km, named CM12) is applied. The timestep length of CM50 is 240 s and of CM12 120 s. Both COSMO-CLM/MESSy instances use 40



vertical model levels (terrain following) with geometric height as the vertical coordinate. The height of the uppermost model level is at $\approx$ 22 km; the damping zone starts at 11 km, the lowest model layer is $\approx$ 20 m thick. The boundary conditions for CM50 are provided by EMAC, the boundary conditions for CM12 are provided by CM50 (Mertens et al., 2020a). To facilitate a one-to-one comparison with observations, EMAC is "nudged" by a Newtonian relaxation of the temperature, the divergence,

the vorticity, and the logarithm of surface pressure (Jöckel et al., 2006) towards ERA5 reanalysis data for the years 2017 to 2019 (Hersbach et al., 2020). Sea surface temperature and sea ice coverage are prescribed as boundary conditions for the simulation set-up from ERA-Interim as well. Due to the MESSy infrastructure, the same diagnostics and chemical process descriptions are applied in all model instances.

## 2.2 Methodology

In this study we use the TAGGING submodel developed by Grewe et al. (2017) and Rieger et al. (2018). The tagging method applies the combinatorical approach described by Grewe (2013). The TAGGING method tags CO, PAN, $O_3$, OH, $HO_2$ and the two families of $NO_y$ and NMHC. The family approach for $NO_y$ and NMHC is chosen due to computational reasons. Detailed definitions of the families are given by Grewe et al. (2017). The source attribution method allows a separation of the emissions by their source and geographical origin. For this, we define four tagging regions: Europe (EU), North America (NA),

East Asia (EA) and the Rest of the World (ROW) to distinguish between $O_3$ from regional sources (i.e. same continent) and from long-range transport (see Figure S1). The notation for tagged ozone in our study is described in detail in Table 1. The regional attribution of the emission fluxes to our tagging regions Europe, North America, East Asia and the rest of the world is enabled by the Simple CALCulations (SCALC) submodel (see Supplement Sect. 3).

The MECO(2) simulation was performed in the QCTM (quasi chemistry-transport model) mode what means that the chem-

istry does not affect the meteorology in global and regional model instances (Deckert et al., 2011). The method ensures the same meteorological conditions, if different emission inventories are used, and is described in detail by Mertens et al. (2016). The usage of the QCTM mode is important for follow up studies with different emission inventories, but not of interest for the present study.

The chemical mechanism used by the submodel MECCA (Module Efficiently Calculating the Chemistry of the Atmosphere)

considers the basic gas-phase chemistry of ozone, methane, odd nitrogen, and other reactants, as described by Sander et al. (2011) and Jöckel et al. (2016). We use the *CCMI2-base-02-tag.bat* mechanisms which is based on Jöckel et al. (2016). It includes basic $NO_x$-CO-$CH_4$-$O_3$ chemistry including the chemistry of isoprene $C_5H_8$ and non-methane hydrocarbons (NMHCs) up to 4 carbon atoms. In comparison to Jöckel et al. (2016) the mechanism has been slightly extended including additional halocarbons and acetonitrile ($CH_3CN$) with OH, $O^1D$ and Cl. The halogen chemistry includes bromine and chlorine species.

Chemistry of sulphur is also considered. The chemical mechanisms of MECCA and the submodel SCAV (scavenging and wet deposition, Tost et al., 2006, 2010), which calculates the scavenging of trace gases by clouds and precipitation are part of the Supplement.



| Tagging category | Description | Notation for tagged ozone |
|---|---|---|
| Land transport ROW | emissions of road traffic, inland navigation, railways (IPCC codes 1A3b_c_e) from Rest of the World | $O_3^{tra}$ |
| Land transport EU | emissions of road traffic, inland navigation, railways (IPCC codes 1A3b_c_e) from Europe | $O_3^{teu}$ |
| Land transport NA | emissions of road traffic, inland navigation, railways (IPCC codes 1A3b_c_e) from North America | $O_3^{tna}$ |
| Land transport EA | emissions of road traffic, inland navigation, railways (IPCC codes 1A3b_c_e) from East Asia | $O_3^{tea}$ |
| Anthropogenic non-traffic ROW | sectors energy, solvents, waste, industries, residential, agriculture from Rest of the World | $O_3^{ind}$ |
| Anthropogenic non-traffic EU | sectors energy, solvents, waste, industries, residential, agriculture from Europe | $O_3^{ieu}$ |
| Anthropogenic non-traffic NA | sectors energy, solvents, waste, industries, residential, agriculture from North America | $O_3^{ina}$ |
| Anthropogenic non-traffic EA | sectors energy, solvents, waste, industries, residential, agriculture from East Asia | $O_3^{iea}$ |
| Shipping | emissions from ships (IPCC code 1A3d) | $O_3^{shp}$ |
| Aviation | emissions from aircraft | $O_3^{air}$ |
| Lightning | lightning-$NO_x$ emissions | $O_3^{lig}$ |
| Biogenic | online calculated isoprene and soil-NOx emissions, offline emissions from biogenic sources and agricultural waste burning (IPCC code 4F) | $O_3^{soi}$ |
| Biomass burning | biomass burning emissions | $O_3^{bio}$ |
| $CH_4$ | degradation of $CH_4$ | $O_3^{CH_4}$ |
| $N_2O$ | degradation of $N_2O$ | $O_3^{N_2O}$ |
| Stratosphere | downward transport from the stratosphere | $O_3^{str}$ |

**Table 1.** Description of the different tagging categories applied in this study following Grewe et al. (2017). Please note that some tagging categories summarize different emission sectors (see description). The last row shows the nomenclature of the tagged tracers for ozone as used in this study. Nomenclature of other species is accordingly.

In accordance to Mertens et al. (2016) we calculate emissions of $NO_x$ by lightning only on the global scale, using the parameterisation by Grewe et al. (2001). In CM50 and CM12 we use the emissions from EMAC (i.e. with same geographical, vertical, and temporal distribution), which are transformed online onto the grids for CM50 and CM12, respectively.

For anthropogenic emissions we applied the EDGAR (Emissions Database for Global Atmospheric Research, version 5.0) inventory for the year 2015 with a monthly time resolution (Crippa et al., 2019b, 2020). The EDGARv5.0 inventory is based on international energy balances, agricultural statistics of the FAO (Food and Agriculture Organization of the United Nations),





and on regional or national assumptions on technology use and emission control standards (Crippa et al., 2019b; Schindlbacher et al., 2021). The dataset is calculated by using a consistent bottom-up approach. The EDGAR data were pre-processed in a way, that the emissions are vertically distributed after Mailler et al. (2013), which is based on Bieser et al. (2011); see Supplement Sect. 4.

The biomass burning emissions are included using the Copernicus Atmosphere Monitoring Service Global Fire Assimilation System (CAMS GFAS, version 1.2) data set from ECMWF (European Centre for Medium-Range Weather Forecasts, Di Giuseppe et al. (2017)). In order to represent the vertical distribution of the wildfire emissions in an appropriate way, the data were pre-processed and vertically distributed onto six height levels after Dentener et al. (2006) depending on the geographical region (see Supplement Sect. 5). Soil $NO_x$ and biogenic isoprene emissions are calculated by the submodel ONline

EMISsions (ONEMIS, Kerkweg et al., 2006)) following the parametrisations of Yienger and Levy II (1995) and Guenther et al. (2006), respectively. Table S1 in the Supplement lists the annual and summer totals of NO emissions in EMAC for the used emission inventories and parametrisations (e.g. lightning). The simulation period for EMAC and CM50 spans from 12/2016 until 02/2019. The finer nested instance CM12 was applied only for the summer months June, July and August (JJA) 2017 and 2018 as well as March and April 2018. These time periods were selected, because during July 2017 and March 2018 two

aircraft measurement campaigns, EMeRGe Europe (see next section) and EMeRGe Asia took place, which provide valuable observation data for the model evaluation.

## 3   Evaluation

During the model evaluation we focus on the results of CM12, as we will use these results mostly for the further analyses. Nevertheless, some evaluations can only be performed with CM50, as longer time periods are needed. Therefore, we compare

in a first step ground-level $O_3$ between CM50 and CM12 for the summer months JJA 2017 and transformed the data of CM12 on the grid of CM50 (Figure S2 in the Supplement). Generally, CM50 and CM12 simulate the same order of magnitude and a similar geographical distribution for ground-level $O_3$.

### 3.1   TOAR dataset

In order to evaluate ground-level $O_3$ simulated by the MECO(n) model in Europe, the CM50 model results are compared with

the data product (in the following D21) created by Delang et al. (2021). The data product contains the seasonal ozone daily maximum 8 h mixing ratio (OSDMA8), which is based on the maximum 8-hourly running mean ozone values (MDA8). They combined a large number of $O_3$ measurements at various stations from the TOAR database with results from nine different atmospheric chemistry models (Delang et al., 2021). The atmospheric chemistry model simulations mostly stem from phase one of the Chemistry-Climate Model Initiative (CCMI). In order to create a multimodel composite of all nine atmospheric

chemistry models, M[3]Fusion was used (Delang et al., 2021). M[3]Fusion corrects the model bias and finds a linear combination of models in each region and year that minimizes the mean square error as compared to observations (Delang et al., 2021). The hourly surface ozone observations cover 1970 to 2015, for 2016 and 2017 fewer data are available compared to 2015. For





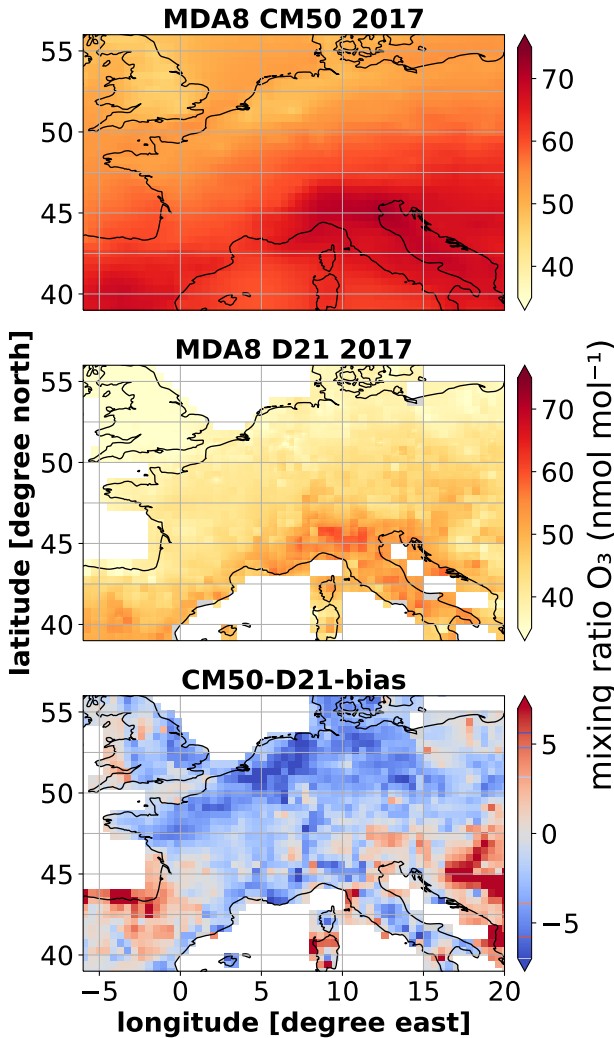

**Figure 1.** Comparison of the 6-month mean of the daily maximum 8-hour ground-level $O_3$ mixing ratio (nmol mol$^{-1}$) for 2017 of CM50 (0.44° x 0.44°) with D21. White areas in D21 are missing values. For the difference of CM50 and D21 the CM50 results have been di-biased as described in the text.

the analyses we first calculated the seasonal ozone daily maximum 8 h mixing ratio (OSDMA8) in CM50 and interpolated the results onto the (coarser) grid of D21. By calculating the area weighted differences between CM50 and D21, ignoring the missing data points, we calculated a mean bias of 16.5 nmol mol$^{-1}$.

Figure 1 displays ground-level OSDMA8 in nmol mol$^{-1}$ for CM50 (upper panel) and the ground-level OSDMA8 in
5   nmol mol$^{-1}$ from the D21 data product in 2017 (middle panel). The lower panel in Figure 1, shows the differences between CM50 and D21 onto the D21 grid. Here, the CM50 results have been reduced by the mean bias of 16.5 nmol mol$^{-1}$. The figure indicates that in CM50 $O_3$ is systematically overestimated with a bias of 16–20 nmol mol$^{-1}$ in rural regions like the Alps,



parts of the Iberian Peninsula, Wales, and the Balkan region. The ozone bias in polluted areas like the Ruhr area, Benelux, parts of France and the Po Valley and in the range of 5–10 $\mathrm{nmol\ mol^{-1}}$.





## 3.2 Ground-level observations

In order to evaluate simulated ground-level $O_3$ concentrations, a comparison with observational data from the AirBase network was performed. AirBase is a European air quality database, maintained by the EEA (European Environment Agency: https://www.eea.europa.eu/data-and-maps/data/aqereporting-8) through its European topic centre on Air pollution and Climate

Change mitigation (Agency, 2018). The database contains air quality monitoring data and information submitted by participating countries throughout Europe. As regional models are typically not able to capture conditions near strong sources (e.g. measurements at traffic sides or near industrial sides), we restrict the evaluation to stations labeled as 'background'. Negative concentrations in the measurement data have been eliminated; missing values are not considered during the evaluation process. In order to compare the model results with the measurements, we sample the model data using a nearest neighbor approach

from hourly model output of CM12.

Figures 2 and 3 show the probability density functions (PDFs) of $NO_x$ and $O_3$ at 253 measurement stations in Europe for July 2017, respectively. The comparison reveals that the frequency of observed $NO_x$ concentrations below 3 µg m$^{-3}$ are overestimated by CM12 (blue line), whereas that for large $NO_x$ concentrations is overestimated. While the frequency of $NO_x$ is underestimated, there is an overall positive bias for the frequency of $O_3$. The model overestimates the frequency of small ozone

values and underestimates the frequency of very large ozone values. The overall mean bias across all stations for $NO_x$ and $O_3$ are -2.8 µg m$^{-3}$ and 19.3 µg m$^{-3}$, respectively. The Root Means Square Errors (RMSE) for $NO_x$ and $O_3$ are 8.9 µg m$^{-3}$ and 31.3 µg m$^{-3}$, respectively. The mean bias and RMSE for ozone is comparable to previous evaluations of MECO(n) (see Table 7 by Mertens et al., 2020b). Reasons for this ozone bias have been discussed in previous publications (Mertens et al., 2016, 2021). One main reason is a too strong vertical mixing during night which leads to too large ozone values. This is is a common

problem in many models (Travis and Jacob, 2019). Main reasons for the underestimations of $NO_x$ are the horizontal resolution of the model, leading to a dilution of emissions over a large area, and uncertainties of the emission inventories.

## 3.3 HALO in situ measurements from EMeRGe Europe

The flight measurement campaign EMeRGe Europe took place in July 2017 with HALO flights across Europe. The goal was to measure emission plumes from major polluted regions and to study their transport and transformation (Andrés Hernández

et al., 2022). Since the focus of our study is on ozone and $NO_y$, the respective in situ measurement data from EMeRGe Europe were used for comparison. A detailed description of the instruments can be found in Andrés Hernández et al. (2022) and Ziereis et al. (2022). In our study, three flights (11.07.2017, 20.07.2017 and 26.07.2017) are analysed, because these flights took place within our study areas (Po Valley and Benelux). In order to facilitate the evaluation of the MECO(n) model with these flight data, the "sampling in 4 dimensions" (S4D) submodel is used. S4D allows to sample on-line the MECO(n) model output along

the flight position at the highest possible frequency, i.e. at every model timestep (Jöckel et al., 2010).

A detailed intercomparison between aircraft in situ measurements and model data is limited. Specific features in the model could be shifted in time (or space) compared to the observations (see also discussion by Andrés Hernández et al., 2022). Therefore, we will here focus on a more qualitative comparison between the measurements and the model results for the



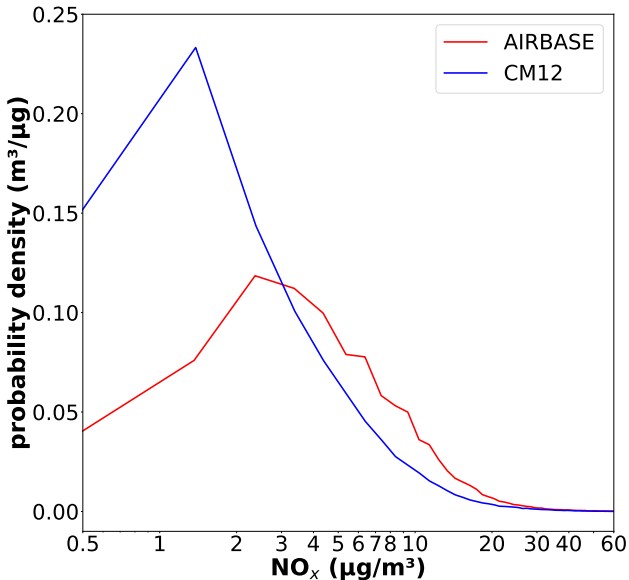

**Figure 2.** Probability density function for July 2017 of the hourly ground-level $NO_x$ concentrations in µg m$^{-3}$ of the model output of CM12 (blue) and the rural AIRBASE station data (red).

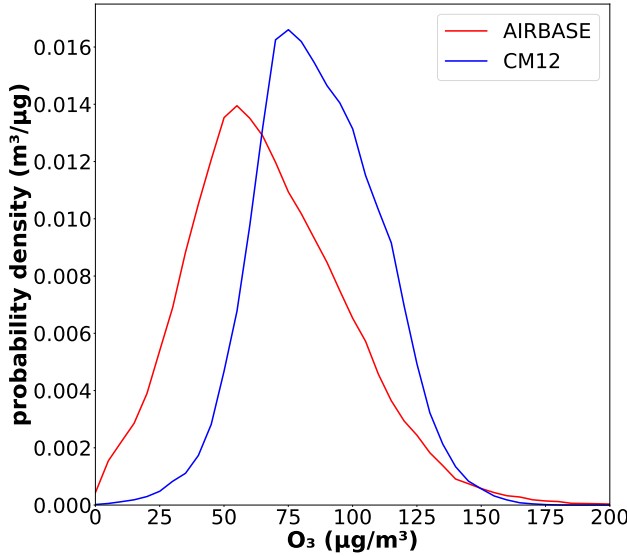

**Figure 3.** Probability density function for July 2017 of the hourly mean ground-level $O_3$ concentrations in µg m$^{-3}$ of the model output of CM12 (blue) and the rural AIRBASE station data (red).

specific flights and regions. Additional scatter plots for the individual flights for a more quantitative intercomparison are shown in the Supplement (Fig. S3 and Fig. S4 in the Supplement). These indicate, that the model performance, compared to the





observations, varies strongly depending on the specific flight. Some show a rather good agreement with observations for $NO_y$ and $O_3$, while other flights (especially the flight on the 11.07.2017) shows a positive ozone bias of 10–15 $nmol\ mol^{-1}$.

Figure 4 depicts the comparison of the observational data with CM12 model results for the Po Valley on the 11th of July 2017. It indicates that $NO_y$ at 725 hPa (left panel) is well represented by the model and observed enhancements of $NO_y$ west

of Genoa are reproduced by the model. The $NO_y$ outflow of the Po Valley west of Venice at 925 hpa agrees geographically and temporally very well (right panel). The vertical profiles of $NO_y$ displayed in Figures 5 and 6 confirm the agreement between measurement and observations.

Compared to $NO_y$, $O_3$ is mostly overestimated, as shown for the pressure levels at 725 hPa and 925 hPa, respectively (Figure 7). This confirms the findings based on the intercomparison with the D21 data set and the ground-level observations. Figures

8 and 9 confirm the overestimation of $O_3$ at 725 hPa in CM12 and show large $O_3$ values above 700 hPa, which has not been measured by HALO. This overestimation is caused by a large scale transport of $O_3$ rich airmasses from France to Northern Italy simulated by MECO(n).

For the flight across the Po Valley, taking place on July 20th, 2017, the simulated geographical distributions of the $NO_y$ plumes at 925 hPa agree with the observations. In this case, however, $NO_y$ is mainly underestimated by the model near city

centres (Milano plume). At the same time, $O_3$ is mainly underestimated by the model near these city plumes, while it agrees well with the observations outside the plumes (see Figs. S5–S8 in the Supplement).

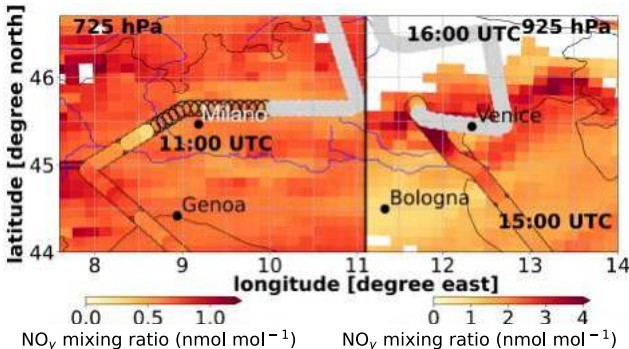

**Figure 4.** $NO_y$ mixing ratios in $nmol\ mol^{-1}$ of the model output of CM12 (background color) at 725 hPa (11 UTC) and 925 hPa (15 UTC) and the HALO in situ measurements (filled circles) for the flight date 11.07.2017 in the Po Valley. The grey filled circles mask the measurement data, where HALO flew above or below the shown pressure level. The white spots mark the grid points in which the surface pressure is lower than 725 hPa (left) and 925 hPa (right), respectively. Unfilled circles mark missing data.



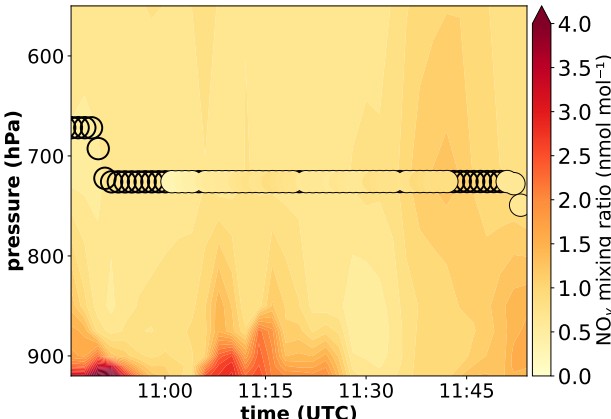

**Figure 5.** Comparison between CM12 model results of $NO_y$ mixing ratios in $nmol\ mol^{-1}$ sampled along the flight path of HALO (background color) with the HALO in situ measurements (filled circles) for the flight date 11.07.2017 in the Po Valley. Unfilled circles mark missing data.

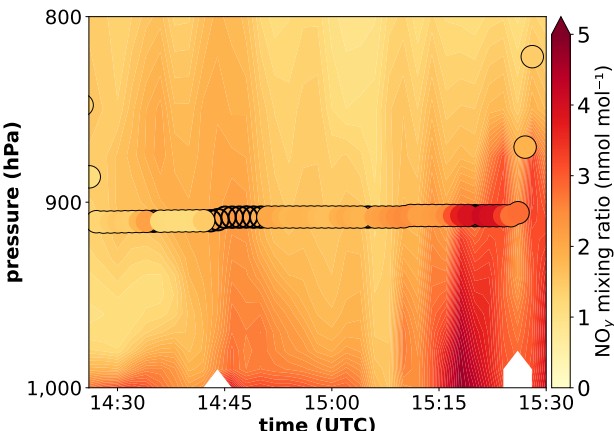

**Figure 6.** Comparison between CM12 model results of $NO_y$ mixing ratios in $nmol\ mol^{-1}$ sampled along the flight path of HALO (background color) with the HALO in situ measurements (filled circles) for the flight date 11.07.2017 in the Po Valley. Unfilled circles mark missing data.



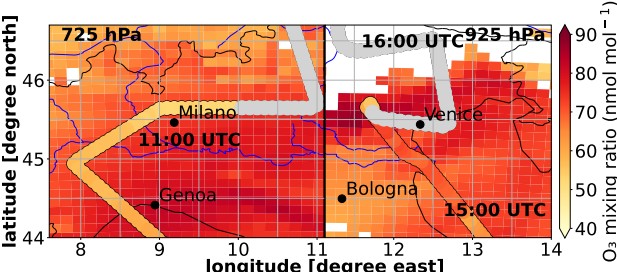

**Figure 7.** $O_3$ mixing ratios in nmol mol$^{-1}$ of the model output of CM12 (background color) at 725 hPa (11 UTC) and 925 hPa (15 UTC) and the HALO in situ measurements (filled circles) for the flight date 11.07.2017 in the Po Valley. The grey filled circles mask the measurement data, where HALO flew above or below the shown pressure level. The white spots mark the grid points in which the surface pressure is lower than 725 hPa (left) and 925 hPa (right), respectively.

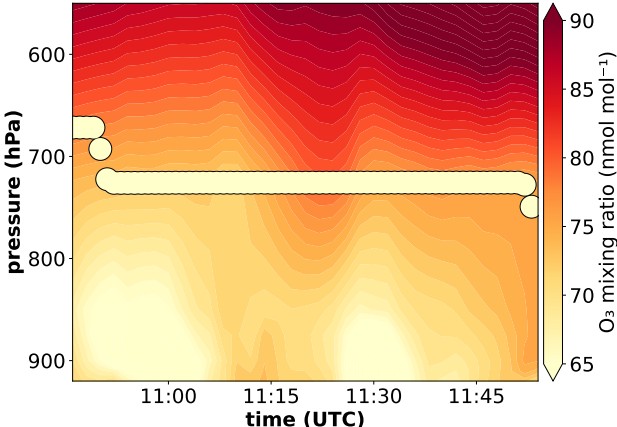

**Figure 8.** Comparison between CM12 model results of $O_3$ mixing ratios in nmol mol$^{-1}$ sampled along the flight path of HALO (background color) with the HALO in situ measurements (filled circles) for the flight date 11.07.2017 in the Po Valley.



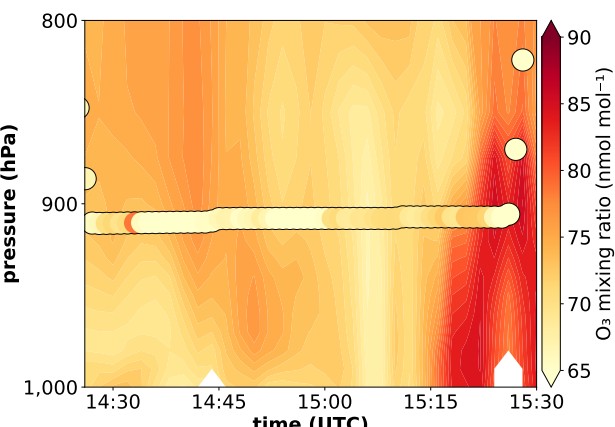

**Figure 9.** Comparison between CM12 model results of $O_3$ mixing ratios in $\mathrm{nmol\ mol^{-1}}$ sampled along the flight path of HALO (background color) with the HALO in situ measurements (filled circles) for the flight date 11.07.2017 in the Po Valley.



Figure 10 displays the measurements from the flight in the Benelux region on July 26th, 2017 in a composite with CM12 data. The $NO_y$ plume of Antwerp is shifted northward in the CM12 simulation results. $NO_y$ is mostly underestimated in the neighborhood of city centres (Figure 11). In between the city plumes, $NO_y$ is well represented by CM12. At the same time $O_3$ is underestimated within plumes, especially between Bruges and Antwerp large $O_3$ mixing ratios are placed too far to the East

5 by CM12 (Fig. 12). Outside the city plume, starting at 12:50 UTC, $O_3$ is very well represented by CM12, which is confirmed by the vertical profiles (Fig. 13). Overall, CM12 is able to capture the variability of $NO_y$ and $O_3$ mixing ratios measured during the aircraft in situ measurements. Specific patterns, however, are shifted in time and space. There is a tendency that in the neighborhood of city centres and in their downwind plumes, the model results partly underestimate $NO_y$ and under-/overestimate $O_3$.

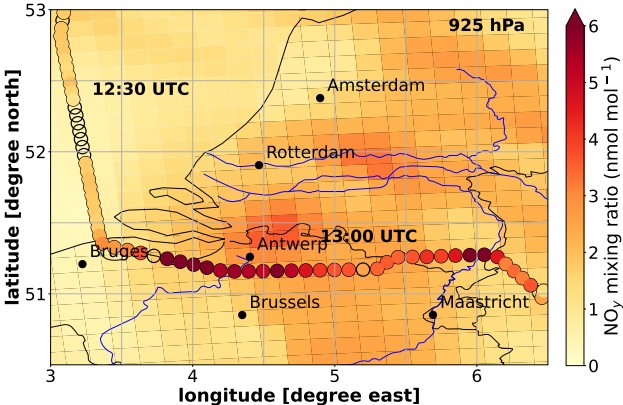

**Figure 10.** $NO_y$ mixing ratios in $nmol\,mol^{-1}$ at 12 UTC of the model output of CM12 (background color) at 925 hPa and the HALO in situ measurements (filled circles) for the flight date 26.07.2017 in the Benelux region. Unfilled circles mark missing data.



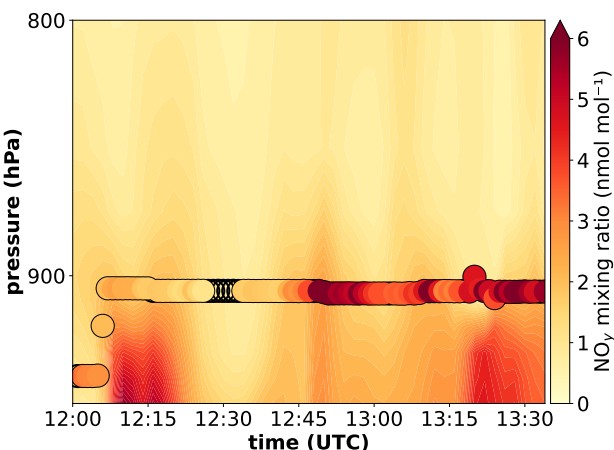

**Figure 11.** Comparison between model results of $NO_y$ mixing ratios in $nmol\ mol^{-1}$ sampled along flight path of CM12 (background color) with the HALO in situ measurements (filled circles) for the flight date 26.07.2017 in the Benelux region. Unfilled circles mark missing data.

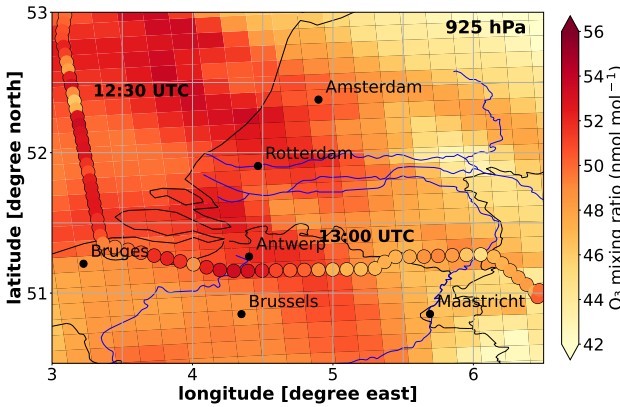

**Figure 12.** $O_3$ mixing ratios in $nmol\ mol^{-1}$ at 13 UTC of the model output of CM12 (background color) at 925 hPa and the HALO in situ measurements (filled circles) for the flight date 26.07.2017 in the Benelux region.





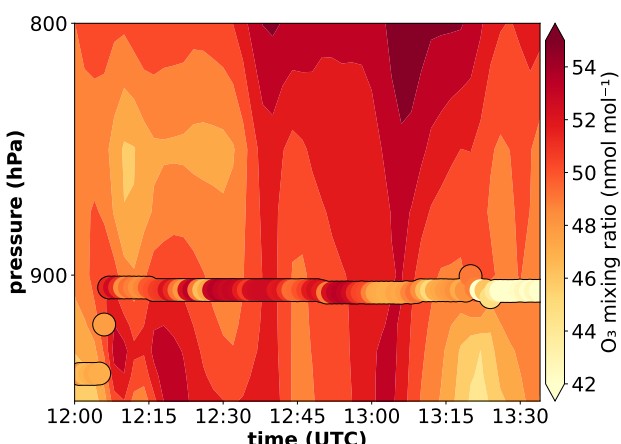

**Figure 13.** Comparison between model results of $O_3$ mixing ratios in $\mathrm{nmol\ mol^{-1}}$ sampled along flight path of CM12 (background color) with the HALO in situ measurements (filled circles) for the flight date 26.07.2017 in the Benelux region.



| Region | Latitude | Longitude | Type of regime |
|---|---|---|---|
| Europe | 33.5° to 56.6° N | 8.3° W to 23.2° E | mixed |
| Po Valley | 45° to 46.5° N | 7° to 14° E | polluted basin |
| Benelux | 50° to 53° N | 3° to 7° E | polluted coastal |
| West Ireland | 51° to 55° N | 8° to 12° W | inflow |
| Spain/Portugal | 37° to 42° N | 4.5° to 8.5° W | rural |

**Table 2.** Definition of the regions, which are analysed in this study in detail. The last column lists the type of the chemical regime of each region.

## 4  Source attribution results

In this section, contributions to ozone from land transport, anthropogenic non-traffic, and biogenic emissions are analysed, because these are the sectors with the largest share to ground-level $O_3$ in Europe (e.g. Karamchandani et al., 2017; Mertens et al., 2018; Butler et al., 2018; Lupaşcu and Butler, 2019; Mertens et al., 2020b). Other emission sectors are either summarized

as rest or not shown/discussed. For our analysis, we define five different study areas with rather large and rather low air pollution in Europe (see Table 2 and Fig. S9 in the Supplement). Besides Europe (whole domain), Po Valley, Benelux, a region in Spain/Portugal (in the following defined as Spain for simplicity) and West Ireland are considered. The latter two are chosen to represent a rural region (Spain) and region which is dominated by Inflow (West Ireland).

In our analyses we focus on July 2017 as the EMeRGe measurement campaign took place in this months. The contributions

for July 2017 are representative for JJA 2017 (Fig. S10 and Fig. S11 in the Supplement). Moreover, contributions for JJA 2017 are similar to JJA 2018 (Fig. S12 and Fig. S13 in the Supplement). As ozone values in summer 20018 are higher than in 2017 (Fig. S14 in the Supplement), absolute contributions in 2018 are generally larger than in 2017, but the relative contributions are similar, with a trend towards larger contributions of ozone from European emissions compared to long-range transport in 2018 compared to 2017.

### 4.1  Contribution of different emission sectors to ground-level ozone

The main focus of this section is to investigate which emission sectors contribute most to ozone values in the considered regions (the region for Spain is omitted in this analyses). Of special importance is the distinction between the geographical origin of the emissions to discriminate between the contributions attributed to long range-transport and to regional emission sources.

As a first step, the contributions of the ozone precursors are analysed in detail. The largest contribution of land transport

and anthropogenic non-traffic emissions to ground-level $NO_y$ in Europe are simulated in the Benelux regions with up to 2–9 nmol mol$^{-1}$ and 2–8 nmol mol$^{-1}$, respectively. The contributions in the Po Valley are around 2–10 nmol mol$^{-1}$ for land transport and 1–4 nmol mol$^{-1}$ from anthropogenic non-traffic (see Fig. S15 in the Supplement). Most of the contribu-



tions to ground-level $NO_y$ in Europe arise from European emissions, only a very small share from long-range transported emissions. For NMHC this is different: here the anthropogenic non-traffic (30–120 nmol mol$^{-1}$) and biogenic sector (up to 30 nmol mol$^{-1}$) are the largest contributors to ground-level NMHC in the Benelux region and the Po Valley. Contributions from land transport emissions are in the range of 3–15 nmol mol$^{-1}$ (Fig. S16 in the Supplement).

Figure 14 shows the absolute contributions to ozone of the various sectors as simulated by CM12 in Central Europe for JJA 2017. The results for CM12 show slightly larger contributions in hot spot regions compared to those of CM50 (Fig. S17 in the Supplement), but the distributions and order of magnitude agree well between CM50 and CM12. In general, the emissions from European anthropogenic non-traffic emissions ($O_3{}^{ieu}$), European land-transport ($O_3{}^{teu}$) and from biogenic emissions ($O_3{}^{soi}$) are the largest contributors to ground-level $O_3$ in Europe. These contributions also show a positive gradient in North-West

to South-East direction. The distribution of the contribution to ozone from long-range transported emissions (anthropogenic non-traffic and land transport) is more homogeneous and largest in South Europe. Reasons for the peak over South Europe are transport of air masses from the African continent (tagged as ROW) to Europe (especially Southern Spain) and descent of air masses transported from North America over the Mediterranean (Stohl et al., 2002; Eckhardt et al., 2004).

The source attribution method yields contributions to ozone of the individual emissions sources and calculates the ozone

production and loss rates for each emission sector, from which we calculate the net ozone production for each emission sector (i) defined as:

$$PO3_i^{net} = ProdO3_i - LossO3_i \qquad (1)$$

Figure 15 shows the total $PO3^{net}$ and $PO3_i^{net}$ for the most important emission sectors (land transport, anthropogenic non-traffic, and biogenic) separated between impact from European emissions (EU) and emissions from other regions (LRT=NA+EA+ROW).

Total $PO3^{net}$ shows a clear North-South gradient, indicating much larger net ozone production in Southern Europe as in Northern Europe. Accordingly, also $PO3^{net}$ in the Po Valley is much larger as in Benelux. Ozone production from European land transport emissions peak in the Po Valley and some larger cities in Southern Europe (Madrid, Rome, Naples). Similarly, also $PO3_i^{net}$ from European anthropogenic non-traffic emissions peak in the Po Valley and around hot-spots mainly in South- and Eastern Europe. Ozone production from biogenic sources is largest over the Iberian Peninsula. In situ production from anthro-

pogenic precursors over Europe from LRT plays almost no role in Europe; only along the ship lanes in the Atlantic ozone production from LRT takes place. This production is due to reactions of $NO_y$ from shipping with NMHC emissions from evaporation of gas/oil transported with ships (not the shipping emissions itself). This NMHC evaporation is categorised as anthropogenic non-traffic emissions from the rest of the world (see Fig. S16 in the Supplement) as they take place over the oceans.

Figure 16 shows the area-averaged contributions to ozone for each study area, except Spain, to ground-level $O_3$ for July 2017. Not explicitly shown sources are summarised as 'others'. A detailed breakdown of the contributions from the 'others' category are given in the Supplement (Figs. S18–S20).





In the Po Valley the absolute ozone contribution from European land transport emissions (red bars) is with around 7 $\mathrm{nmol\,mol^{-1}}$ larger than in the Benelux region with around 3 $\mathrm{nmol\,mol^{-1}}$. Similarly, absolute contributions from European anthropogenic non-traffic emissions (orange bars) are larger in the Po Valley (11 $\mathrm{nmol\,mol^{-1}}$) than in Benelux (5 $\mathrm{nmol\,mol^{-1}}$). This is consistent with the differences of the net ozone production rates between both regions stated above. In contrast, $NO_y$ contributions from European land transport emissions and European anthropogenic non-traffic emissions are larger in the Benelux region than in the Po Valley (Fig. S15 and Fig. S16 in the Supplement). In contrast to this, Ireland shows almost no ozone contributions from European anthropogenic sources. Instead, shipping, biogenic (which is a global source) and others (mainly biomass burning and $CH_4$) dominate ozone in the inflow region (see Fig. S20).

Ozone contributions from global shipping emissions are similar in the Benelux region and the Po Valley with around 3 $\mathrm{nmol\,mol^{-1}}$. Absolute ozone contributions from global biogenic emissions ($O_3^{soi}$) are largest in the Po Valley with more than 12 $\mathrm{nmol\,mol^{-1}}$, and smaller in the Benelux region with 6 $\mathrm{nmol\,mol^{-1}}$. Accordingly, ozone contributions from $O_3^{soi}$ in the Po Valley are twice as large as in the Benelux region. This agrees with the ozone net production in the Po Valley, which is up to 3 times larger than in the Benelux region (Figure 15, lower left panel). While in the Benelux region soil-$NO_x$ emissions are larger than in the Po Valley, biogenic emissions of isoprene are larger in the Po Valley as in the Benelux region (see Fig. S21 and Fig. S22 in the Supplement). Since our tagging mechanism combines different ozone precursors, no clear statement can be made if $O_3^{soi}$ arises mostly from soil-$NO_x$ or isoprene emissions in each region.

The absolute contributions to ground-level ozone of sources summarized as others in the Po Valley are 15 $\mathrm{nmol\,mol^{-1}}$ in the Po Valley and 10 $\mathrm{nmol\,mol^{-1}}$ in the Benelux region (Fig. 16). In both regions the relative contributions of the sectors summarised as others are similar with the most important contributions from $CH_4$, biomass burning and lightning-$NO_x$ (see Figs. S18–S20 in the Supplement).

The relative ozone contributions of $O_3^{teu}$ and $O_3^{ieu}$ in the Po Valley are 12 % and 19 %, respectively (Fig. 17). In accordance with the absolute contributions and the larger net ozone production in the Po Valley compared to Benelux also these relative contributions are larger than in Benelux (7 % $O_3^{teu}$ and 15 % $O_3^{ieu}$). The relative ozone contributions for the sum of land transport emissions from other regions are slightly smaller with 5 % in the Po Valley than in the Benelux region with 6 %. The emissions from land transport in North-America contribute with 3 % and 4 % similarly to ground-level $O_3$ in the Po Valley and in the Benelux region. East Asian land transport emissions contribute only 1 % to ground-level $O_3$ in both regions, because transport times are similar or larger to the lifetimes of the ozone precursors. Furthermore, air masses from East Asia are also strongly diluted and mixed with other emissions during the transport. In the Po Valley, the relative ozone contributions from anthropogenic non-traffic emissions from other regions of the world are 9 %, in more detail 3 % originate from NA, 4 % from ROW and 2 % from EA. In the Benelux region, even larger relative ozone contributions of anthropogenic non-traffic emissions from other regions are simulated (13 %). Here, the relative contributions to ozone of ROW and NA are slightly smaller as in the Po Valley with 5 % and 5 %, respectively. The remaining part (3 %) comes from East Asia. This distribution is favoured by the coastal location and the large-scale weather pattern of the Benelux region, which is mostly dominated by fronts. During the intercontinental transport from NA, ozone and precursors are diluted during the advection across the Atlantic. This leads to lower mixing ratios of long-range transported $NO_y$ and $O_3$ arising from North-America, and therefore to a uniform distribution



across Europe (Figure 14). The relative contribution from biogenic emissions to ozone is 22 % in the Po Valley and 19 % in the Benelux. Accordingly, biogenic emissions are one of the most important contributor to ground-level $O_3$.

The relative contribution from shipping emissions to ground-level $O_3$ is around 9 % in Benelux which is around twice as much as in the Po Valley ( 5 %). This is due to the coastal location and strong influence of the shipping emissions over the North Sea. West Ireland has even slightly larger relative ozone contributions from shipping emissions compared to Benelux, because of the proximity to important Atlantic shipping routes.

Figure 17 shows that in all three study areas the sum of all other non-anthropogenic sectors (blue bar) contribute to ozone by 27–37 %. $O_3^{CH_4}$ and $O_3^{bio}$ are the largest contributions with 7–12 % and 5-11 % to ground-level $O_3$, respectively (see Fig. S18–S20 in the Supplement). $O_3^{lig}$ and $O_3^{str}$ are also very important sectors to ground-level $O_3$ and their relative contribution to ground-level $O_3$ is quite uniformly distributed across Europe with 7 % and 5 %, respectively.

In Benelux and the Po Valley, ozone contributions from European anthropogenic emissions (land transport + non-traffic) have the largest ozone share with 18 $\mathrm{nmol\,mol^{-1}}$ in the Po Valley and 7 $\mathrm{nmol\,mol^{-1}}$ in the Benelux region. This comparison shows that the mitigation potential for the European anthropogenic sector in the Benelux region is more limited than in the Po Valley, because much less ozone is produced in situ from regional emissions. Instead, ozone is more dominated by long-range transport and shipping in Benelux compared to the Po Valley.





**Figure 14.** Monthly mean absolute contribution as mixing ratios in nmol mol$^{-1}$ of O$_3$ for JJA 2017 from long-range transported (LRT: ROW + NA + EA), biogenic, and European land transport as well as anth. non-traffic emissions as simulated with CM12.



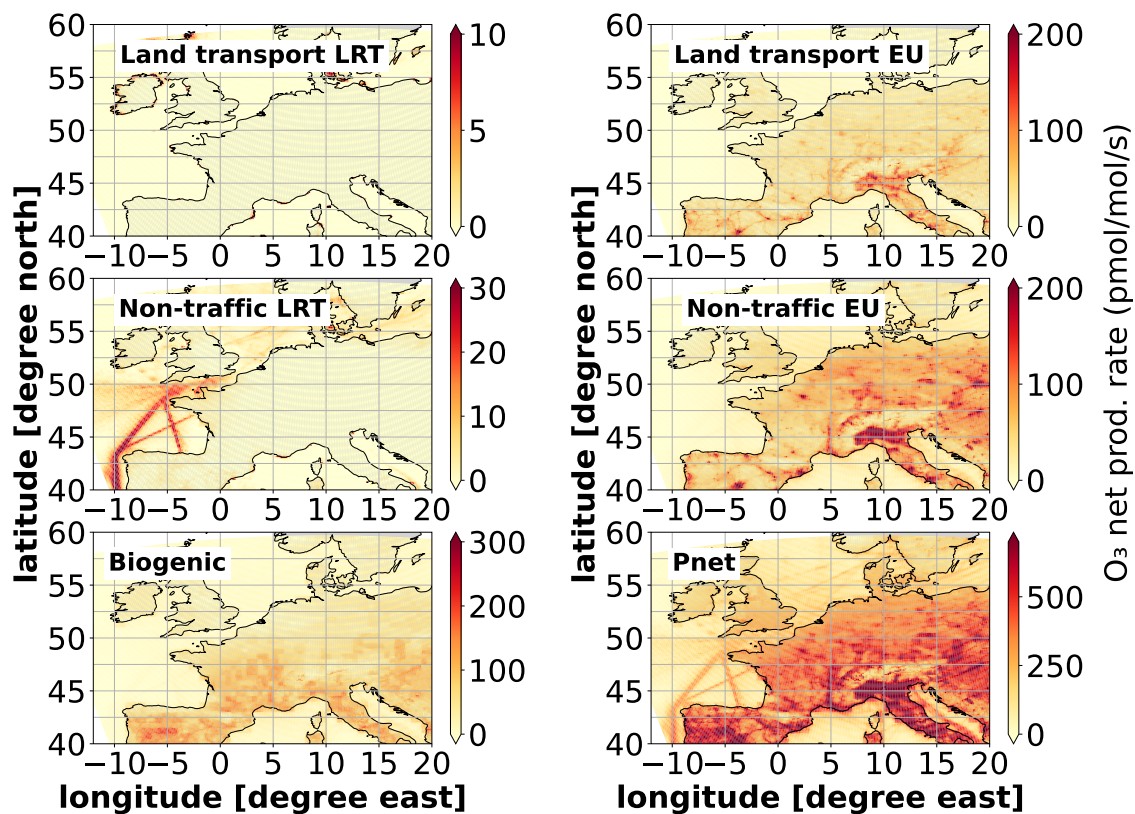

**Figure 15.** Seasonal mean (JJA 2017) of the ozone net production ($P_{net}$=Production-Loss; see chemical mechanism in Supplement: ProdO$_3$ and LossO$_3$) in $\mathrm{pmol\ mol^{-1}s^{-1}}$ from long-range transported (LRT: ROW + NA + EA), biogenic, and European anthropogenic non-traffic and land transport emissions.



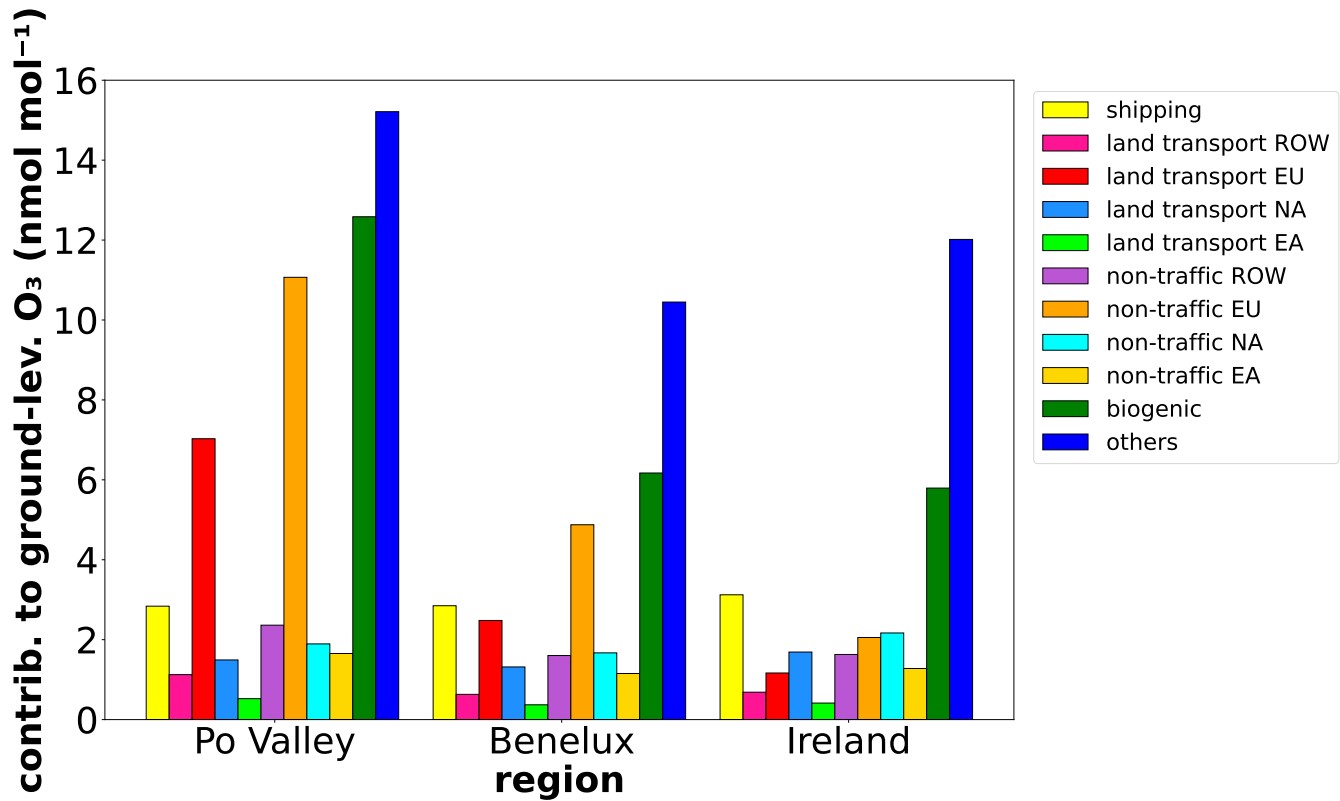

**Figure 16.** Monthly mean absolute contribution of different emissions sectors and regions to ground-level ozone in the three regions Benelux, Po Valley, and West Ireland for July 2017 as simulated with CM12.



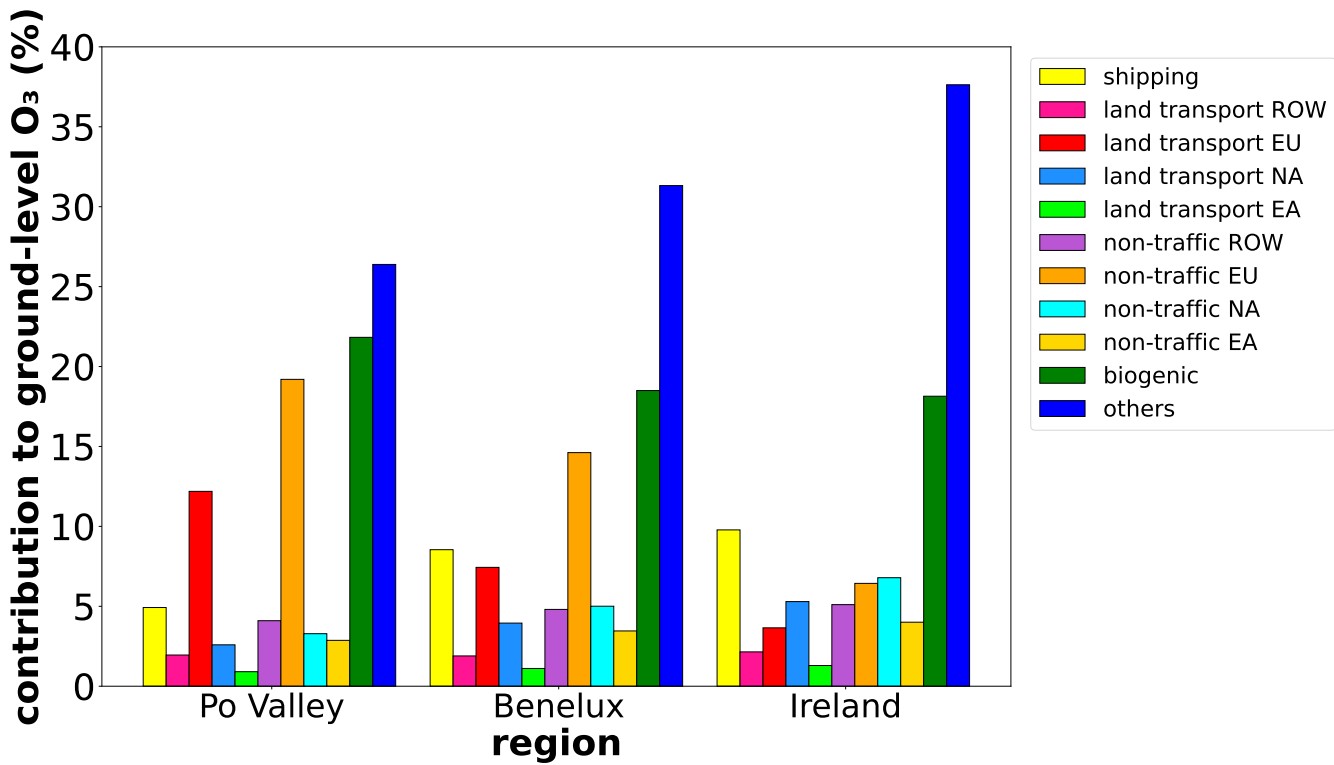

**Figure 17.** Monthly mean relative contribution of different emissions sectors and regions to ground-level ozone in the three regions Benelux, Po Valley, and West Ireland for July 2017 as simulated with CM12.





## 4.2 Contributions during periods of large ozone values

Especially for human health, periods of large ozone values are most harmful. Such large ozone values can occur, for example, under stagnant conditions during heat waves. During these periods contributions to ozone can differ strongly from seasonal mean values (e.g. Mertens et al., 2020b; Lupaşcu et al., 2022). Therefore, we investigate the contributions at the 95th, 90th and 75th percentiles of ozone. The analyses are performed for whole Europe and for the four study-regions (see Table 2).

Figure 18 shows the absolute contributions of land transport, anthropogenic non-traffic and biogenic emissions at different percentiles of ozone (see Fig. S23 in the Supplement for relative contributions). As the ozone values can have a large geographical spread, the analysis are presented as box-whisker plots to indicate the spatial variability of contributions among the regions. The results show that contributions of $O_3^{\mathrm{ieu}}$ (1–22 nmol mol$^{-1}$, 2–35 %)) and $O_3^{\mathrm{teu}}$ (1–13 nmol mol$^{-1}$, 2–20%) have a large geographical variability over Europe, while the variability of the contributions from long range transport are much smaller (compare also Fig. 14). The large spread is favoured by an in-homogeneous distribution of the emission sources together with a strong variability of the net ozone production.

In accordance with Mertens et al. (2020b) contributions from land transport, anthropogenic non-traffic and biogenic sources increase in the regions Europe, Benelux, Po Valley and Spain with increasing ozone percentiles. Compared to Mertens et al. (2020b) our additional information about the geographical origins of the emissions shows that the increase of the contributions of land transport and anthropogenic emissions is caused by emissions from within Europe. Here, absolute contributions from European land transport emissions increase to up to 16 nmol mol$^{-1}$ (20 %) for the 95th percentile of ozone. Contributions of European anthropogenic non-traffic emissions increase up to 25 nmol mol$^{-1}$ (35 %) at the 95th percentile of ozone.

The contributions of both emission sectors from long range transport remain relatively constant at all ozone percentiles. In accordance with Fig. 14 the difference between contributions from long range transport and from European emissions is largest in the Po Valley, where the net ozone production is also largest. Accordingly, large ozone values can be reduced very well in the Po Valley by reduction of European emissions. Compared to this, the differences between contributions from long range transport and European emissions is much smaller in Benelux and Spain. In both regions the contributions from European emissions is only slightly larger as the contribution from long range transport (especially at the 95th percentile). Larger contributions from German emissions at large ozone levels in Germany have also been reported by Lupaşcu et al. (2022). Our results indicate that the overall potential to reduce large ozone values by reduction of European emissions is much smaller in Benelux and Spain compared to the Po Valley. This is also in accordance with contribution analyses of peak ozone values over the Iberian Peninsula by Pay et al. (2019).

Compared to Po Valley, Benelux and Spain the contributions from long range transport and European emissions are very similar in Ireland. In addition, the contributions from land transport and anthropogenic non-traffic long range transport emissions show no increase for increasing ozone percentiles. The contributions from biogenic emissions, however, show a small increase.

An important metric for ozone exceedences is MDA8, therefore we also investigate ozone contributions to MDA8. To do so, we first calculated MDA8 for July 2017. Based on the MDA8 values we first calculated mean, minimum and maximum MDA8



values for July 2017 (geographical distributions are given in Fig. S9 in the Supplement). For these maximum, minimum and mean values of MDA8 the contributions are analysed for the study regions defined in Table 2. Similar as for the percentiles, the contributions are analysed as box-whisker plots to indicate the geographical spread among the regions in Fig. 19. Please note, for a better comparability of the anthropogenic emissions with biogenic emissions, the contributions of the anthropogenic

emissions in Fig. 19 are the sum of contributions from all regions (i.e. for land transport $O_3^{teu}$+ $O_3^{tna}$+$O_3^{tra}$+$O_3^{tea}$). A figure with the same analysis differing between contributions from European emissions and long range transport is part of the Supplement (Fig. S24).

The monthly maxima of MDA8 over the whole domain range from 40–100 nmol mol$^{-1}$ (Fig. 19). Contributions to the maximum MDA8 of land transport emissions (sum of all regions) range between 4–22 nmol mol$^{-1}$, contributions from an-

thropogenic non-traffic (sum of all regions) between 9–37 nmol mol$^{-1}$ and contributions from biogenic emissions between 5–30 nmol mol$^{-1}$. The analysis largely confirm the findings from the analysis of the percentiles. From minimum to maximum MDA8 values the absolute contributions of land transport, anthropogenic non-traffic and biogenic emissions increase over Benelux, Po Valley and Spain. Over the Po Valley large MDA8 values are strongly driven by European emissions, while over Benelux and Spain also long-range transport plays a role (see Fig. S24 in the Supplement). For Ireland the minimum, maximum

and mean MDA8 values change only slightly, and also the corresponding contributions of the three emission sectors are nearly constant independent of MDA8.





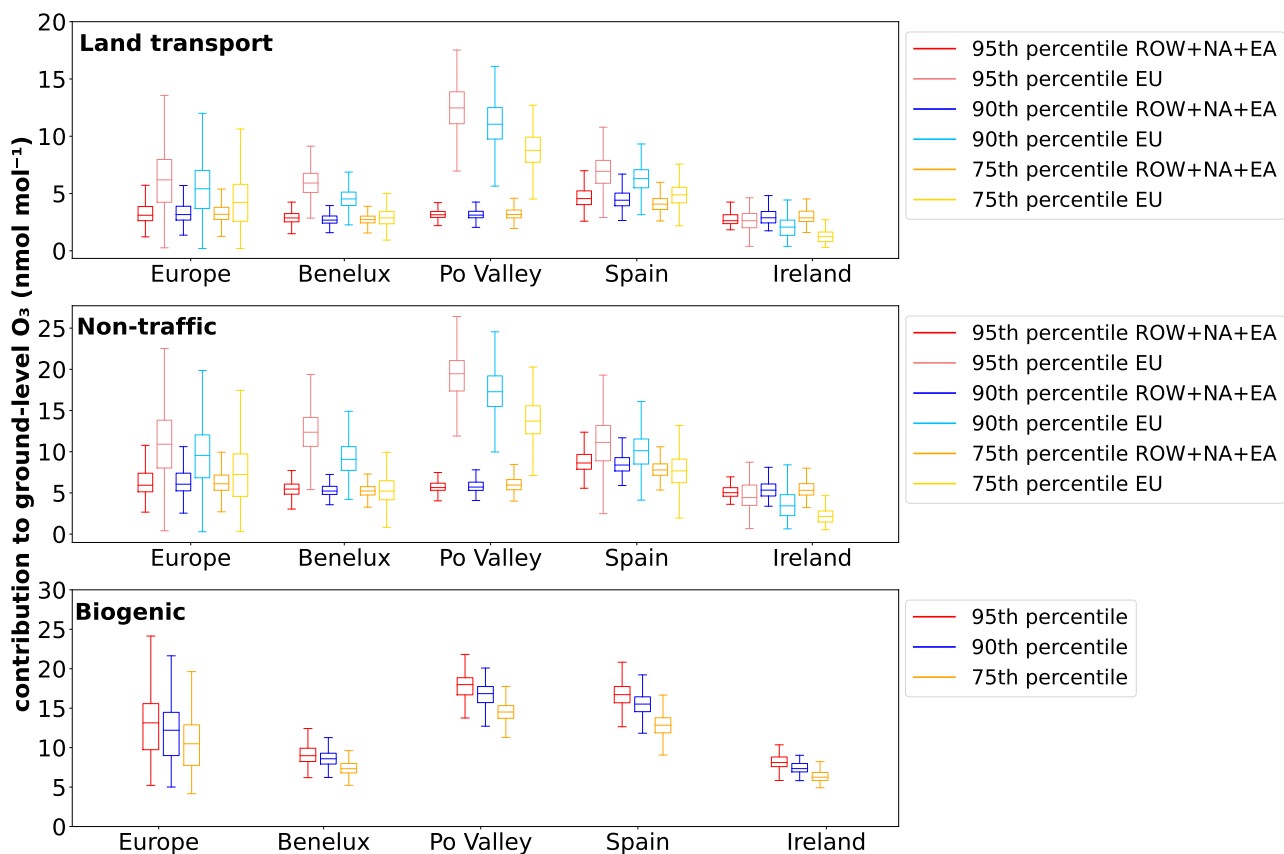

**Figure 18.** Box-whisker plot showing the contributions (in $\mathrm{nmol\ mol^{-1}}$) of the most important emission sources for the 95th, 90th, and 75th percentiles of ozone as simulated by CM12 for July 2017. The first panel shows the regional absolute contributions of $O_3^{\mathrm{teu}}$ (labeled EU) and the sum of long-range transported absolute contributions of $O_3^{\mathrm{tra}}$, $O_3^{\mathrm{tna}}$ and $O_3^{\mathrm{tea}}$ (labeled ROW+NA+EA). The second panel shows the absolute contributions of $O_3^{\mathrm{ieu}}$ (labeled EU) and the sum of long-range transported absolute contributions of $O_3^{\mathrm{ind}}$, $O_3^{\mathrm{ina}}$ and $O_3^{\mathrm{iea}}$ (labled ROW+NA+EA). The third panel shows the absolute contributions of $O_3^{\mathrm{soi}}$. The lower and upper ends of the boxes indicate the 25th and 75th percentile of the corresponding regional distribution, respectively, the bar the median, and the whiskers are defined as $\pm 1.5$ the interquartile range of the contributions of all grid boxes within the indicated region.





**Figure 19.** Box-whisker plot showing the contributions in $\mathrm{nmol\,mol^{-1}}$ of the most important emission sources of ozone as simulated by CM12 for July 2017. Shown are ozone and the contributions of land transport, anthropogenic non-traffic and biogenic emissions to ground-level ozone during the monthly maximum of the maximum daily 8-h average (MDA8), the monthly mean of MDA8 and the monthly minimum of MDA8 based on 1-hourly model output. The lower and upper ends of the boxes indicate the 25th and 75th percentile corresponding regional distribution, respectively, the bar the median, and the whiskers are defined as $\pm 1.5$ the interquartile range of the contributions of all grid boxes within the indicated region.



## 5    Discussion

Emissions of $NO_x$ and VOCs are important precursors for the formation of $O_3$. As shown, emissions from land transport and anthropogenic non-traffic sources are important for $NO_x$ (here analysed as $NO_y$) over Europe, while for VOCs especially biogenic and anthropogenic non-traffic sources are most important. The regions with the largest contribution of land transport and anthropogenic non-traffic emissions to $O_3$, however, are not always identical with the regions of the largest contributions to $NO_x$ and $NO_y$. Mertens et al. (2020a) already stated that large amounts of $NO_x$ emissions do not necessarily lead to large $O_3$ mixing ratios. The reason is the non-linearity of the ozone chemistry, depending on the availability of VOCs (Sillman, 1999), and the strong dependence of the ozone mixing ratio on the meteorological conditions affecting deposition and transport (Vieno et al., 2010; Francis et al., 2011; Logan, 1985).

The analysed contributions, however, depend on the applied emission inventories. The model results and the observational data of $NO_x$ and $NO_y$ show a good agreement, but locally, $NO_y$ is underestimated (Sect. 3). The applied anthropogenic emission inventory (EDGAR5.0) relies on various assumptions (Crippa et al., 2020). The precursor emissions are subject to considerable uncertainties (Schindlbacher et al., 2021). Moreover, anthropogenic and biomass burning emissions, as well as biogenic and natural emissions are uncertain and the range of uncertainties is difficult to quantify (Davidson and Kingerlee, 1997; Yienger and Levy II, 1995; Vinken et al., 2014). Our results must be viewed in the light of these uncertainties. Considering the discussed studies, however, we expect that results change only slightly as long as the ratio of the emission strengths and the order of magnitude of specific emissions do not change drastically. To quantify the impact of different emissions in more detail, we compare the results of our study with Mertens et al. (2020b). They used the same model and tagging method, but different emission inventories on global and regional scale. In addition, Mertens et al. (2020b) considered the years 2008–2010 and did not consider different geographical regions. Therefore, we summed up the contributions from different geographical region in our study to compare the values roughly with Mertens et al. (2020b). This comparison is summarised in Table 3 indicating that the results are relative robust with respect to different emissions and years.

The source attribution method applied in this study also comes with some simplification, as discussed by Grewe et al. (2017). As Mertens et al. (2020a) already clarified, the mathematical method itself is accurate, but the implementation into the model requires some simplifications such as the introduction of chemical families (e.g. $NO_y$, NMHC). These simplification can lead to small artifical contribuions. Such an example is NMHC from the lightning category, which is created from decomposition of PAN from lightning into $NO_y$ and NMHC. PAN from lightning is created from reactions of $NO_y$ from lightning with NMHC from other emission categories (Grewe et al., 2017; Butler et al., 2018). In order to quantify the influence of simplifications and also the influence of different tagging approaches onto the source attribution results we further compare our results with results from other source attribution studies.

First of all we compare our results with publications using the tagging approach described by Butler et al. (2018). In this approach ozone is attributed - in different simulations - either to emissions of $NO_x$ or VOCs, while our approach considers $NO_x$ and VOCs at the same time. Sources with large amounts of $NO_x$ emissions but only low VOC emissions, like shipping or lightning-$NO_x$, end up in larger contributions in the $NO_x$ tagging approach of Butler et al. (2018). This has been discussed



**Table 3.** Comparison of absolute and relative contributions for the sectors land transport, anthropogenic non-traffic, and biogenic between Mertens et al. (2020b) and this study. Please note that , the regions and sector definitions are not identical between the two studies. Regions and sectors therefore corresponds to the definitions in our study. Values for Benelux by Mertens et al. (2020b) correspond to their values for Mid Europe, values for Spain by Mertens et al. (2020b) correspond to their values for the Iberian Peninsula, values for West Ireland correspond to their Inflow region, and the values for anthropogenic non-traffic from Mertens et al. (2020b) also contains shipping and aviation. All values are for JJA (2017 in this study; 2008–2010 in Mertens et al. (2020b)).

| Sector | Po Valley | Benelux | Spain | Inflow |
|---|---|---|---|---|
| | | this study | | |
| land transport | 16 – 17 % | 12 % | 14–15 % | 11 – 12 % |
| land transport | 9–10 nmol mol$^{-1}$ | 5 nmol mol$^{-1}$ | 8–9 nmol mol$^{-1}$ | 4 nmol mol$^{-1}$ |
| | | Mertens et al. (2020b) | | |
| land transport | 14 – 16 % | 13–14 % | 11–12 % | 8–9 % |
| land transport | 7–9 nmol mol$^{-1}$ | 5–6 nmol mol$^{-1}$ | 6 nmol mol$^{-1}$ | 4 nmol mol$^{-1}$ |
| | | this study | | |
| anth. non-traffic | 30–31 % | 28–30 % | 24–25 % | 25–27 % |
| anth. non-traffic | 17–19 nmol mol$^{-1}$ | 11–13 nmol mol$^{-1}$ | 14–15 nmol mol$^{-1}$ | 9–10 nmol mol$^{-1}$ |
| | | Mertens et al. (2020b) | | |
| anth. non-traffic | 27–31 % | 28–32 % | 30–32 % | 33–34 % |
| anth. non-traffic | 14–17 nmol mol$^{-1}$ | 12–14 nmol mol$^{-1}$ | 16–18 nmol mol$^{-1}$ | 15 nmol mol$^{-1}$ |
| | | this study | | |
| biogenic | 22 % | 17 % | 20 % | 16 % |
| biogenic | 13 nmol mol$^{-1}$ | 7 nmol mol$^{-1}$ | 11–12 nmol mol$^{-1}$ | 6 nmol mol$^{-1}$ |
| | | Mertens et al. (2020b) | | |
| biogenic | 19–20 % | 19–20 % | 18–19 % | 14 % |
| biogenic | 9–10 nmol mol$^{-1}$ | 8–9 nmol mol$^{-1}$ | 10–11 nmol mol$^{-1}$ | 6–7 nmol mol$^{-1}$ |

already by Butler et al. (2020) for shipping emissions. Interestingly, Butler et al. (2020) report contributions of lightning-$NO_x$ for the receptor region North-West Europe mixing ratios of below 1 nmol mol$^{-1}$ during JJA 2010 which is lower as the 4 nmol mol$^{-1}$ we find for our Benelux region in July 2017. Even though, the definition of the receptor regions differs, it can not explain such a large difference. Another reason, besides different tagging approaches, could be different years (and therefore

meteorology) and lightning-$NO_x$ emissions. The lightning-$NO_x$ emissions, however, are comparable between the two studies (Butler et al. (2020) around 3.43 Tg(N) a$^{-1}$ while we apply 3.68 Tg(N) a$^{-1}$), but also the geographical distribution might differ. Another explanation could also be differences in vertical mixing, as also the ozone attributed to the stratosphere in Butler et al. (2020) is smaller ($\approx$ 1 nmol mol$^{-1}$) compared to 4 nmol mol$^{-1}$ in our study. For biogenic contributions, containing soil-$NO_x$ and biogenic VOCs, our results are in between the results of the $NO_x$ and VOC tagging simulations. As example, Butler

et al. (2018) report ground-level ozone contributions in Europe in July 2010 from biogenic $NO_x$ emissions of 5–7 nmol mol$^{-1}$



and 15–25 nmol mol$^{-1}$ from biogenic VOC emissions. The results from our global model (see Fig. S25 in the Supplement) are in the range of 6–16 nmol mol$^{-1}$. Similarly, Butler et al. (2020) reported ground-level ozone contributions from soil NO$_x$ emissions of 4–5 nmol mol$^{-1}$, and from biogenic VOC emissions of 10–13 nmol mol$^{-1}$. Since we are not able to distinguish between ozone contributions from NO$_x$ and NMHC emissions, our results for Europe (July 2017) are in the range of 7–22 5 nmol mol$^{-1}$ and thus comparable.

Finally, Butler et al. (2018) diagnosed contributions from anthropogenic NO$_x$ sources in the range of 25–35 nmol mol$^{-1}$ with a positive North-West to South-East gradient across Europe. In our global model instance contributions in the range of 15–35 nmol mol$^{-1}$ (see Fig. S26 in the Supplement) are calculated with a similar gradient. In accordance, Butler et al. (2020) report ground-level ozone contributions from Northwest European (local) NO$_x$ emissions of 10–13 nmol mol$^{-1}$ in JJA 2010, 10 which is larger than our results of 8 nmol mol$^{-1}$ for the Benelux region.

Lupaşcu and Butler (2019) applied the Butler et al. (2018) tagging method within the WRF-Chem model (Weather Research and Forecasting (WRF) model coupled with Chemistry model) that attributes O$_3$ concentrations in several European receptor regions to NO$_x$ emissions. They calculated in their GEN (Germany, Belgium, Netherlands, and Luxembourg) region means of regional contribution from anthropogenic (land transport and non-traffic) emissions for July to September 2010 of 25 % (their 15 Fig. 7). Compared to this, we find a contribution of 23 % from anthropogenic emissions from Europe in the Benelux region for JJA 2017.

In summary, as discussed in previous studies (Butler et al., 2020), our approach of tagging NO$_x$ and VOC concurrently provides lower contributions from anthropogenic sources, which have mainly NO$_x$ but not that much VOC emissions, compared to a seperate tagging of NO$_x$ or VOCs.

Other tagging methods, often applied on the regional scale (Dunker et al., 2002; Kwok et al., 2015) check if the ozone chemistry is either NO$_x$ oder VOC limited and attributes ozone production accordingly either to NO$_x$ or VOC. Karamchandani et al. (2017) applied such a source attribution method in the Comprehensive Air quality Model with Extensions (CAMx) and estimated the ozone contribution from road transport emissions for different metropolitan regions across Europe. They found contributions to MDA8 during summer of 19 % in Amsterdam and 11 % in London. Our results for Amsterdam are 9 % and 25 7–8 % for London and therefore lower in both cases. With a similar approach Pay et al. (2019) used the source attribution method with the CMAQ model and estimated a relative ozone contribution from on-road transport emissions of 11–16 % in most parts of Spain during summer, which corresponds to our results (13–17%) quite well (Fig. S27 in the Supplement).

In summary, a direct comparison of results from different studies with different tagging approaches is limited. Different approaches (e.g. tagging NO$_x$ and VOCs separately or at the same time) can lead to larger differences between the contributions 30 of specific sources. Generally, however, the different approaches give contributions of similar magnitude. In detail however, different emission inventories and different models make it hard to quantify the differences due to different tagging approaches. Therefore, a detailed intercomparison would be very valuable to quantify differences caused by different tagging approaches.

A limitation of our study is the choice of the three large tagging regions which are large scale and represent almost entire continents. Thus, no country-wise source-receptor relationships can be established. Therefore, the tagging regions could be further 35 redefined within Europe or in other regions (e.g. country by country), in order to enable the source attribution to each coun-





try (e.g. by Lupaşcu and Butler, 2019). This implementation is mainly limited by the amount of memory per computing task and the low I/O speed during reading emissions for many regions which slows down the computational speed tremendously. Nevertheless, it could be worthwhile for the assessment of mitigation strategies on national scale.

## 6 Conclusions

In the present study we investigate the ozone contributions of anthropogenic (land transport and non-traffic) and biogenic emissions in Europe. By means of simulations with the MECO(n) model system we analysed contributions in several regions in detail. The model system allows an online coupling of the global chemistry–climate model EMAC with the regional chemistry–climate model COSMO-CLM/MESSy. In order to quantify the contribution of land transport, anthropogenic non-traffic, and biogenic emissions to ozone and it's precursors, a tagging method for source attribution is used. To distinguish between
regional and long-range transported contributions, we define four different geographical source regions, Europe, North America, East Asia, and the Rest of the World. Our tagging method fully decomposes the budgets of ozone and ozone precursors into contributions from various emission sources (and regions) and is applied in the global and regional model instances. Here, we focus on monthly mean contributions to ground-level ozone during the summer 2017. The simulated ozone contributions from European anthropogenic emissions ($O_3^{teu}$ + $O_3^{ieu}$) and from biogenic emissions ($O_3^{soi}$) are the largest contributors to
ground-level $O_3$ during summer in Europe with a positive gradient in North-West to South-East direction. The spatial distribution of the ozone contribution from long-range transported emissions is more homogeneous and has its largest absolute values in South Europe.

The first goal was to assess the differences of the $NO_y$ and $O_3$ contributions by various emission sectors between two polluted areas, the Benelux region and the Po Valley. Contributions of land transport and anthropogenic non-traffic emissions
to ground-level $NO_y$ for JJA 2017 are 2–9 nmol mol$^{-1}$ and 2–8 nmol mol$^{-1}$ in the Benelux region, respectively. In the Po Valley, the simulated contributions from land transport and non-traffic emissions to ground-level $NO_y$ are smaller with 2–10 nmol mol$^{-1}$ and 1–4 nmol mol$^{-1}$, respectively. Ozone contributions for JJA 2017 in the Po Valley from land transport and anthropogenic non-traffic emissions are 10–13 nmol mol$^{-1}$ (18–20 %) and 17–21 nmol mol$^{-1}$ (30–32 %), respectively. For the Benelux region these ozone contributions are smaller with 5 nmol mol$^{-1}$ (13–15 %) and 8–11 nmol mol$^{-1}$ (27–30 %),
respectively.

The second goal was, to quantify the contributions from European emissions in comparison to the contributions from long-range transported emissions to ground-level $O_3$. In the Po Valley the geographically averaged ozone contributions for JJA 2017 from European land transport ($O_3^{teu}$) and anthropogenic non-traffic emissions ($O_3^{ieu}$) are 7 nmol mol$^{-1}$ and 11 nmol mol$^{-1}$, respectively. These contributions are more than twice as large as in the Benelux region ($\approx$ 3 and 5 nmol mol$^{-1}$). Even though,
emissions of $NO_y$ are larger over Benelux than in the Po Valley the net ozone production in the Po Valley is much larger. These large $O_3$ formation rates are favoured by warmer temperature, more stagnant conditions and larger insulation.

The relative $O_3$ contributions show similar results. For European land transport and anthropogenic non-traffic emissions the contributions in the Po Valley are 12 % and 21 %, respectively, while the contributions in the Benelux region are 7 % and 15 %,



respectively. In the Po Valley, the relative contribution to ozone from long-range transported land transport emissions (i.e. $O_3{}^{tna}$ + $O_3{}^{tea}$ + $O_3{}^{tra}$) to ground-level $O_3$ is 5 %, and thus slightly lower than in the Benelux region with 6 %. North American and East Asian land transport emissions hardly contribute to ground-level $O_3$ in the Benelux region and the Po Valley, because those emissions have the longest way to Europe, the precursor lifetime is shorter than the transport time and emissions are diluted during the transport. This leads to lower mixing ratios of long-range transported $NO_y$ and $O_3$, as well as to a nearly uniform distribution throughout Europe.

The ozone production rates show the importance of the biogenic emissions ($O_3{}^{soi}$), which are one of the largest contributors to ozone in Europe. In the Po Valley, $O_3{}^{soi}$ is twice as large as in the Benelux region.

The geographical variability of the ozone contribution from European land transport and non-traffic emissions at the 95th percentile is very large throughout Europe. This is favoured by an inhomogeneous distribution of the emission sources, which is less inhomogeneous in rural regions like Spain and West Ireland. Our analysis of the ozone contributions indicate that high ozone events are mainly caused by European emissions from land transport and anthropogenic non-traffic sources, whereas precursors from distant sources dominate the contribution to ozone during low ozone periods. Especially in hot spot regions like the Po Valley and the Benelux region, large ozone mixing ratios are mainly caused by large ozone contributions from land transport and anthropogenic non-traffic emissions.

In the Benelux region and in Spain the geographical variability of the monthly maximum of MDA8 is around one third smaller than in the Po Valley. For West Ireland we simulate lower ozone mixing ratios and nearly no geographical variability of the monthly maximum of MDA8, which implies a rather homogeneous distribution of ground-level $O_3$ in areas, where distant precursor emissions dominate the contribution to $O_3$. In the Po Valley, ozone contributions from European anthropogenic emissions (sum of land transport and anthropogenic non-traffic) during the monthly maximum of MDA8 cause up to 58 nmol mol$^{-1}$ of the monthly maximum of the MDA8 of more than 100 nmol mol$^{-1}$. The same also applies to the Benelux region, where ozone contributions from European anthropogenic emissions contribute up to 50 nmol mol$^{-1}$ to the monthly maximum of MDA8 of 70 nmol mol$^{-1}$. In both regions, almost one half of the ground-level ozone is caused by European anthropogenic emissions, but the absolute ozone contributions are larger in the Po Valley and therefore the mitigation potential to reduce extreme ozone events is larger there, compared in the Benelux region.

For subsequent studies, we intend to further refine the source regions for the apportionment (tagging) in order to allow a country-by-country attribution (e.g. as by Lupaşcu and Butler, 2019). This is a prerequisite to assess the potential benefit of national mitigation regulations.



*Acknowledgements.* This study was supported by the DLR transport program (projects Data and Model-based Solutions for the Transformation of Mobility – DATAMOST – and Transport and Climate – TraK) and by the DLR impulse project ELK (EmissionsLandKarte). PJ acknowledges support from the Initiative and Networking Fund of the Helmholtz Association through the "Advanced Earth System Modelling Capacity (ESM)" project. We acknowledge the use of the tool cdo (https://code.zmaw.de/projects/cdo) for the processing of data. Further, we acknowledge the EDGAR 5.0 emission data (Crippa et al., 2019a) set available at https://edgar.jrc.ec.europa.eu/dataset_ghg50 (last access, 13.03.2023). In addition, we are very thankful to the CLM-Community (clm-community.eu, last access 13.03.2023) for providing and maintaining the COSMO-CLM model. For our study we used resources of the Deutsches Klimarechenzentrum (DKRZ) granted by its Scientific Steering Committee (WLA) under project ID bd1063. We thank Heidi Huntrieser for very valuable comments that improved the manuscript.

## 6.1 Data and Code availability

The Modular Earth Submodel System (MESSy) is continuously further developed and applied by a consortium of institutions. The usage of MESSy and access to the source code is licenced to all affiliates of institutions which are members of the MESSy Consortium. Institutions can become a member of the MESSy Consortium by signing the MESSy Memorandum of Understanding. More information can be found on the MESSy Consortium Website (http://www.messy-interface.org). The code presented here has been based on MESSy version d2.55.2-1913 and will be available in the next official release (2.56). The simulation results used here are archived at the German Climate Computing Center (DKRZ) and are available on request.

## 6.2 Author Contributions

MK, MM and PJ built the model set up and performed the simulations at the DKRZ. AK co-developed MECO(n) and helped during the study with model updates, bug fixes and definition of the model set-up. MK analysed the results and drafted the manuscript. VG supported the interpretation of the source attribution results. HZ and AZ provided the in situ data from the EMeRGe Europe campaign. All authors contributed to the interpretation of the results and to the revision of the manuscript.

## 6.3 Competing interests

The authors declare that they have no conflict of interest.



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
