# Peer review of "Ozone source attribution in polluted European areas during summer as simulated with MECO(n)"

_EGUsphere, 2023_

## Community Comment (CC1)

Dear Referee #1,

Thanks a lot for your review. We would like to reply to a few of your remarks and will answer in more detail (including specific changes to the manuscript) during the peer-review process.

In the following, your statements are given *in italic*, our reply is given in red.

*The authors apply an ozone tagging method in a global chemistry-climate model to attribute the origin of surface ozone pollution in Europe with focus on the Po Valley and the Benelux.*

Please note that we do not apply a global chemistry-climate model only, but an on-line coupled global-regional chemistry-climate model. This allows us to study air pollution in detail on the regional scale considering also global impacts. In detail, we present the results of the regional instances (focusing on the results with 10 km horizontal resolution) in the manuscript.

*The work is carefully done but it's not clear to me that there is anything new in the methods or results. I felt that I was reading a technical report rather than a scientific paper, with a tedious deluge of numbers and figures but no real new insight about the origin of ozone. The source attribution for ozone is consistent with what has been documented in many previous papers.*

We would like thank you for pointing this out. While writing the publication we indeed might be lost in some details. Obviously, we have not highlighted the novelty of our work in detail. Therefore, we would like to highlight our novel approach:

- We apply an on-line coupled global-regional chemistry climate-model with a combined $NO_y$-/VOC source attribution (tagging) and separate four different source regions. Moreover, we distinguish 10 source categories and put emphasize on the role of traffic emissions.

New insights / Highlight:

- Here we find that the contributions from individual sources which have large $NO_x$ but rather few VOC emissions are lower, if their emissions of NOx and VOCs are regarded concurrently.

By design some of our results differ from previous source attribution studies using a $NO_x$-or VOC tagging only. We discuss this in detail in Sect. 5. Given the novel approach we think the work adds additional information to the topic and most importantly confirms previous findings with a different methodology.

*The model is not particularly successful at reproducing observations, so it's not clear to me that the source attribution here deserves any more confidence than previous studies. I don't think that this paper is up to the scientific standards of ACP.*

We don't agree with this statement. As mentioned above the methodology is different from previous publications. If we find comparable results, this increases the confidence of the scientific community of our understanding of the contribution of regional/global sources to ozone; it is a piece in the big puzzle of the tropospheric ozone budget.

*Maybe I'm missing new scientific insights coming from the paper because they are not properly advertised and/or buried. I couldn't find them in the abstract. I would suggest that*

*the authors submit a much shorter paper focused on what is scientifically new in their results, and including proper citation to the literature.*

We will highlight the novelty of our results in more detail in a revised version and will also state this more prominent in the abstract.

1. *The introduction discusses at length the difference between perturbation analysis and attribution by tagging. This is an old story and I don't find it particularly interesting. There's nothing wrong with tagging, it just shouldn't be interpreted as a linear response to a perturbation, and we can leave it at that. It would seem more appropriate for the intro to review past relevant studies on attribution of ozone pollution in Europe - this is lacking.*

We agree that the difference between Tagging and Perturbation is an "old story", however from discussions during the review process of previous publications and from discussions on conferences etc. we have the feeling that there is still a lack of understanding of the differences in large parts of the scientific community. Therefore, we would like to highlight the differences again. If the editor/referee #2 agrees with your opinion we are of course happy to rewrite and shorten the introduction.

During the peer-review process we will provide a more detailed answer including also an overview about changes to the manuscript.

Best regards,

Mariano Mertens (on behalf of all authors)

---

## Author Comment (AC1)

Dear referee#2,
thank you very much for your in-depth review of our manuscript acp-2019-715. Please find our replies to your comments below. Your original comments are repeated in italics, our replies in normal font, and text passages which we included in the text are in bold. Please note, due to the significant changes in the manuscript so many changes have been added that we do not include all revised texts to the reply but refer to the ranges in our diff document.

*This manuscript describes an attribution of summertime ozone over Europe to anthropogenic, natural and transport sources using a tagging approach in the MECO(n) model. The study is competently performed and described and the results are potentially useful for regional ozone mitigation efforts. However, the paper is largely descriptive rather than analytical, and does not provide much new insight into either the source contributions or their broader context, and this reduces its value greatly. While the paper should ultimately be appropriate for publication, it is not suitable in its current form. The results require a deeper level of interpretation to explain their consequences and why they matter, and the paper needs a clearer and more distinct message (preferably demonstrating some originality) that sets it apart from previous studies and makes it worthy of publication.*

Answer: We thank referee #2 for the constructive comments to improve our manuscript. In order to reduce the "descriptiveness" of the manuscript, and in view that referee #1 asked for a shorter manuscript, we decided to move large parts of the model evaluation into the supplementary material and to focus on the attribution results.

As we replied to referee #1, in the revision we more clearly elaborate on the new aspects of our manuscript in comparison to earlier studies, in particular:

- We apply an on-line nested global/regional CCM to account for finer (12 km) resolution in the target area, but also consider consistently the effect of long range transport.

- The attribution is for $NO_x$ **and** NMHCs **concurrently**.

- With our attribution we distinguish four different source regions **and** 10 sectors.

- The focus of our analyses is on the (land) transport sector.

- Besides JJA mean contributions, we also focus on MDA8 ozone.

Please find the changes in the manuscript on the following pages of our diff document:

- page 1 - page 2, l39; revised abstract

- page 3, l72 - page 6, l162 ; revised introduction

*The quantification of the different sources over Europe is interesting, and the focus on two important but contrasting regions of Europe is valuable. The focus on a range of metrics, including the upper tail of the ozone distribution and the responses of MDA8 are particularly valuable and are to be commended.*

Answer: Thank you very much for your appreciation.

*However, the results are not fully exploited, and a greater degree of interpretation is required in the text about why the regions differ and what the consequences of this are.*

This is indeed a very good point, and in a former version we had included some discussion about the chemical regimes in the different regions, but finally we skipped that for the sake of a shorter manuscript. In light of the shortened evaluation we add a new Section 6 with a short analysis on how/why the regions differ. Moreover, we replaced Figure 19 of the original manuscript in the revised version with a more comprehensive version which distinguishes also between long-range transport and European emissions for the MDA8 analysis.

*What are the implications of the attribution for mitigation efforts, are past source changes evident in recent ozone trends (given the attribution that has been derived), and what are the likely contributions to future ozone changes?*

Answer: We are afraid that these interesting questions are far beyond the scope of the current manuscript. The analysis of trends, mitigation efforts of future ozone change would require combined attribution/perturbation simulations as performed by Mertens et al. (2024). Since we did not perform such simulations here, we can't answers these (indeed) very important questions. However, this is planned for follow-up studies.

*The treatment of uncertainty in the paper is weak. Contributions are typically given in the form of large ranges (whether this is spatial or temporal variance is not clear) for a single summer with a single emissions inventory. How sensitive are the results likely to be to changes in meteorology or to the reliability of the emissions used? Quantification (or at least estimation) of these uncertainties would give the reader greater confidence in the results presented.*

Answer: In our discussion we refer to Mertens et al. (2020), who discuss the (large) uncertainty w.r.t. emission inventories. Further, we note that the results for 2018 are similar as for 2017 and show corresponding analyses in the Supplement, implying a minor influence of inter-annual variations in the same climate state. We chose the summer 2017 in the manuscript as during this summer the EMeRGe-Europe campaign took place, of which the data we could use for evaluation of the simulation results. Furthermore, we focus on summer, because we

are particularly interested in large ozone events. We clarify this in the revised text.

The box whisker plots always show the variance with respect to the geographical variability. Also all other ranges refer to the geographical variability. We clarify this in the manuscript. Moreover, the meaning of the shown variances are clarified in the revised figure captions by adding the following note:

The lower and upper ends of the boxes indicate the 25th and 75th percentiles, respectively, the bar the median, and the whiskers are defined as $\pm 1.5$ the inter-quartile range **of all grid boxes within the geographical region.**

*There is some comparison of results with previous tagging studies, although this is unsatisfying given the differing approaches used; how do the results compare with other estimates of source contributions based on observational estimates or non-tagging approaches?*

Answer: The differing approaches are indeed an issue for the inter-comparison of various results, however, as mentioned in the reply to referee #1, our analysis shows that the $NO_x$ only tagging gives larger contributions for sources with large $NO_x$ but small VOC emissions compared to our approach. This clearly demands for a dedicated inter-comparison study using different approaches with different models, however, under the same boundary conditions. This is clearly beyond the scope of our study. Moreover, we are not aware of any observation based estimates of ozone source attribution. In case the referee knows about such studies, we are very interested in learning about those.
Given that perturbation and tagging approaches answer different scientific questions and the derived potential impacts and the contributions can not be compared directly, we are hesitant to add a detailed comparison to these studies in our manuscript. Especially as referee #1 noted that 'tagging' vs 'perturbation' in an 'old story'.

*The discussion somewhat undermines the application of tagging approaches for source attribution through highlighting the differing attribution to NOx or VOC sources dependent on the method used (e.g., comparison with the Butler approach). Under these circumstances, the perturbation approach appears more useful, as it provides a clear, unique ozone response to an applied change.*

Answer: As discussed in many previous studies (and as summarised in our introduction), the tagging / source attribution method does not provide any information about an ozone response to a perturbation of one (or various) precursor emissions. Results of both methods are per-se not comparable, because both answer different questions. Since the literature is already full with discussions about that, we are very much hesitating to repeat the fundamental concepts here in detail again, and we rather refer to the literature we cite in our manuscript. Moreover, we would also like to stress that also the same perturbation yields very different responses in different models (e.g. Fink et al., 2023).

*The conclusions need sharpening up. The current text summarizes results from each part of the analysis, but does not bring them together well to generate a clear and coherent message from the study. The summary of results needs to be cut back (by about half?) and some synthesis of findings should be added.*

Answer: Good point. We revised the conclusion considerably. You can find the revised conclusion on:

- page 34, l726 - page 36, l800; of the diff document.

**Specific Comments**

*The aims are clearly stated in the introduction, but I would like to see a stronger statement about why the study matters. It is valuable to understand how ozone may change in contrasting parts of Europe in future under different mitigation scenarios, but this isn't stated as a motivation. Why is source attribution important?*

As mentioned above and in numerous earlier studies, with the attribution method alone, we do not have the possibility to estimate **changes** under different mitigation scenarios. This can only be achieved by combined attribution/perturbation simulations. This is, however, not in the scope of our study. Here, with the attribution method alone, the focus is on understanding the fundamentals of the underlying processes, in particular the shares of different sectors and regions in ozone for one specific state (or realisation) of the atmospheric composition. In that sense, these shares provide some preliminary hints on mitigation potentials, however, due to the non-linear responses of ozone chemistry, **not** the effects of mitigation measures. The latter can only be achieved with the perturbation approach. Even though, referee #1 asked for skipping this (repeatedly in literature occurring) explanation, we decided to revise large parts of the introduction to state our aim of the study in more detail.

You can find the revised introduction on page 3, l72 - page 6, l162 of the diff document.

*P.6: Why is lightning neglected for the regional models? This may have a non-negligible impact on source attribution in summer, particularly in southern Europe, so some justification is needed.*

Answer: This is a misunderstanding. We do not neglect lightning, we rather calculate lightning NOx only in the global model instance, but we map these results onto the nested regional model instances, in order to achieve exactly the same production rates in all model instances. This is clarified in the revised manuscript by adding the following sentence:

**This approach allows us to use the same amount and the same spatial/temporal distribution of the lightning $NO_x$ emissions in all model instances.**

*Fig 1: Subtraction of the mean bias in panel (c) is potentially misleading and not physically meaningful; I suggest that the full bias is plotted, with an appropriate monochromatic (not dichromatic) color scale that emphasizes the key features of interest.*

Answer: We think that the information on spatial variation gets masked with a (biased) monochromatic scale. But, in the course of the revision, in particular for the sake of the requested shortening of the manuscript by referee #1, this figure is moved to the supplementary material.

*Page 9: What do we learn from the TOAR evaluation? There are substantial biases in the model simulations; while this is true for most models, the reader needs to understand why this is the case and how this is likely to impact the source attribution generated in the study. The same is also true for the HALO evaluation.*

Answer: As the referee states, such biases are common for most comparable models, and we would be happy to understand the exact reasons for it. As we discuss at the end of the evaluation section one main reason is enhanced vertical mixing, especially during night. In consequence, simulated contributions from the stratosphere, from lightning, and from $N_2O$ decomposition to ground level ozone are likely biased high. We added this at the end of the evaluation section: **In consequence, simulated contributions from the stratosphere, from lightning, and from $N_2O$ decomposition to ground level ozone are likely biased high.**

*Figs 5 and 6 should be combined; so should Figs 8 and 9. Please consider combining Figs 4-9 into just two figures, either by species (Fig 4/5/6, Fig 7/8/9) or, better, by orientation (Fig 4/7, Fig 5/6/8/9). This would help the reader get a clearer overview of the comparisons. Similarly Figs 10-13 should be combined into a single figure. Using different colors (consistently) for NOy and O3 would aid the reader in interpretation.*

Answer: These figures are moved to the supplementary material, nevertheless we grouped them as suggested.

*Page 19, line 3: roughly what contribution do these sectors make?*

Answer: In the revised text a rough estimate is provided:
**In more detail, Mertens et al. (2020) report contributions of these sectors during summer of up to 16 % (land transport), 20 % (biogenic) and 30 % (anth. non-traffic).**

*Page 19, lines 9-11: show the key results first, before referring to the supplementary results.*

Answer: Indeed, this is a bad style and changed in the revised manuscript.

*Page 19, line 17: Why was Spain omitted? A short explanation is needed.*

Answer: Good point. Spain is added in the revised manuscript.

*Fig 14: Please use the same color scale for the contribution plots (a-e) so that the reader can compare the contributions directly.*

Thanks for that point. We revised the plots as suggested.

*Figs 16-17: These would be clearer if colors were chosen to provide harmonization across a specific sector (land transport, non-traffic) or region (ROW,EU,NA,EA) using similar hue but contrasting saturation (for example). The same is true for Figs S10-S13 in the Supplement.*

Thanks a lot also for this comment. We revised the plots as suggested.

*Page 21, line 14: Figs S21 and S22 are useful, but it would help the reader to quantify the differences in soil NOx and biogenic isoprene emissions in the text, e.g., by providing regional JJA average fluxes over each region.*

Answer: We added a Table S2 in the revised Supplement and refer to the Table in the manuscript.

*Page 21, line 17: remind the reader what is included in the "others" category*

Answer: Thanks. In order to shorten the manuscript and to make the text less descriptive, this text has been deleted from the revised manuscript.

*Page 27, line 1: "high ozone concentrations" would be clearer in the title here (and in the text) than "large ozone values".*

Answer: "Concentrations" would be wrong, but we change it into "large ozone mixing ratios" in the revised text, and we avoid "high" because it can be misinterpreted with referring to altitude.

*Figure 18: For direct comparison of these panels, it would be useful to have the same scale on the Y-axis (0-25 or 0-30 ppb). This is also true for the three lower panels in Figure 19, where the scale could be 0-40 ppb.*

Answer: Thanks. It is revised accordingly.

*Page 31, lines 12-13: "various assumptions", "considerable uncertainties": please be specific here. The discussion of uncertainties here is weak and qualitative, and a more thorough and quantitative assessment is needed here.*

Answer: Thanks for pointing this our. We revised the text accordingly. The new text reads:

**The uncertainty estimate for a previous version (4.3.2) of the EDGAR emissions by Crippa et al. (2018) indicate uncertainties of $NO_x$ emissions of $17-69$ % depending on the country. For EU-28 in 2012 (the most recent year covered in that analysis), uncertainties of 51 % are reported. Besides the estimates of anthropogenic emissions, also estimates of biogenic and natural emissions are uncertain, as example estimates of the emissions of $NO_x$ from soil range from 4 to 15 $\mathrm{Tg\ (N)\ a^{-1}}$ (Vinken et al., 2014), and emissions from lightning-$NO_x$ range from 2 to 7 $\mathrm{Tg\ (N)\ a^{-1}}$ (Schumann and Huntrieser, 2007).**

*Page 33, lines 28-32: this paragraph undermines the study by casting doubt on the value of the results. A more quantitative approach to tackling uncertainties would allow these issues to be addressed, and would provide more confidence for the reader on the value of the results presented.*
Answer: Since we provide a very detailed and quantitative discussion and comparison with results of other studies in the text above, we removed this paragraph from the revised text, since it was only meant to show overall limitations in a very general sense (which might be obvious, indeed).

*Page 34, lines 1-2: It would be clearer to say that the approach adopted here is not practical for use with a large number of regions.*

Answer: No, we do not agree. The approach **is** possible, however costly w.r.t. computational resources. We reformulated the sentence accordingly.

*Page 34, lines 32-33: combine this with previous paragraph (the topic is the same)*

Answer: We have changes the overall paragraph in the revised version.

*Typos and minor issues There are a relatively large number of typographical errors that need to be cleaned up.*

Answer: Thanks four pointing us to this mistakes. All recommended changes have been applied, unless they have become obsolete due to the overall revision

**References**

M. Crippa, D. Guizzardi, M. Muntean, E. Schaaf, F. Dentener, J. A. van Aardenne, S. Monni, U. Doering, J. G. J. Olivier, V. Pagliari, and G. Janssens-Maenhout. Gridded emissions of air pollutants for the period 1970–2012 within edgar v4.3.2. *Earth System Science Data*, 10(4):1987–2013, 2018. doi: 10.5194/essd-10-1987-2018. URL https://essd.copernicus.org/articles/10/1987/2018/.

L. Fink, M. Karl, V. Matthias, S. Oppo, R. Kranenburg, J. Kuenen, J. Moldanova, S. Jutterström, J.-P. Jalkanen, and E. Majamäki. Potential impact of shipping on air pollution in the mediterranean region – a multimodel evaluation: comparison of photooxidants $no_2$ and $o_3$. *Atmospheric Chemistry and Physics*, 23(3):1825–1862, 2023. doi: 10.5194/acp-23-1825-2023. URL https://acp.copernicus.org/articles/23/1825/2023/.

M. Mertens, A. Kerkweg, V. Grewe, P. Jöckel, and R. Sausen. Attributing ozone and its precursors to land transport emissions in europe and germany. *Atmospheric Chemistry and Physics*, 20(13):7843–7873, 2020. doi: 10.5194/acp-20-7843-2020.

M. Mertens, S. Brinkop, P. Graf, V. Grewe, J. Hendricks, P. Jöckel, A. Lanteri, S. Matthes, V. S. Rieger, M. Righi, and R. N. Thor. The contribution of transport emissions to ozone mixing ratios and methane lifetime in 2015 and 2050 in the shared socioeconomic pathways (ssps). *EGUsphere*, 2024:1–45, 2024. doi: 10.5194/egusphere-2024-324. URL https://egusphere.copernicus.org/preprints/2024/egusphere-2024-324/.

U. Schumann and H. Huntrieser. The global lightning-induced nitrogen oxides source. *Atmospheric Chemistry and Physics*, 7(14):3823–3907, 2007. doi: 10.5194/acp-7-3823-2007.

G. C. M. Vinken, K. F. Boersma, J. D. Maasakkers, M. Adon, and R. V. Martin. Worldwide biogenic soil $no_x$ emissions inferred from omi $no_2$ observations. *Atmospheric Chemistry and Physics*, 14(18):10363–10381, 2014. doi: 10.5194/acp-14-10363-2014.

---

## Author Comment (AC3)

Dear referee#1,
we thank you very much for your in-dpeth review of our manuscript acp-2019-715. Please find our replies to your comments below. Your original comments are repeated in italics, our replies in normal font, and text passages which we included in the text are in bold. Please note, due to the significant changes in the manuscript so many changes have been added that we don't include all revised texts to the reply but refer to the ranges in our diff document.

*The authors apply an ozone tagging method in a global chemistry-climate model to attribute the origin of surface ozone pollution in Europe with focus on the Po Valley and the Benelux.*

Answer: Please note that we do not apply a global chemistry-climate model only, but an on-line coupled global-regional chemistry-climate model. This allows us to study air pollution in more detail on the regional scale considering also global impacts (e.g. by long-range transport). The focus of the manuscript is on the results of the nested regional model instances, i.e. on the results with 12 km horizontal resolution.

*The work is carefully done but it's not clear to me that there is anything new in the methods or results. I felt that I was reading a technical report rather than a scientific paper, with a tedious deluge of numbers and figures but no real new insight about the origin of ozone. The source attribution for ozone is consistent with what has been documented in many previous papers. The model is not particularly successful at reproducing observations, so it's not clear to me that the source attribution here deserves any more confidence than previous studies. I don't think that this paper is up to the scientific standards of ACP. Maybe I'm missing new scientific insights coming from the paper because they are not properly advertised and/or buried. I couldn't find them in the abstract. I would suggest that the authors submit a much shorter paper focused on what is scientifically new in their results, and including proper citation to the literature.*

Answer: Thank you very much for pointing this out. While writing the publication, we indeed might have lost ourselves in some details. Obviously, we have not highlighted the novelty of our work in sufficient detail. Therefore, we would like to highlight our novel approach:

- We apply an on-line nested global/regional CCM to account for finer (12 km) resolution in the target area, but also consider consistently the effect of long range transport.

- The attribution is for $NO_x$ **and** NMHCs **concurrently**.

- With our attribution we distinguish four different source regions **and** 10 sectors.

- The focus of our analyses is on the (land) transport sector.

- Besides JJA mean contributions, we also focus on MDA8 ozone.

New insights / highlight:

- We find that the contributions to ozone from individual sectors, which have large NOx but rather few VOC emissions, are estimated to be lower, if their emissions of $NO_x$ and VOCs are regarded concurrently (in comparison to studies which attribute either only $NO_x$ or only VOCs).

  By design, some of our results differ from previous source attribution studies using a NOx or VOC tagging only. We discuss this in detail in Sect. 7. Given the novel approach, we think our study adds additional information to the topic and not least confirms previous findings with a different methodology.

To highlight the novelty in more detail in the revised manuscript, we changed the introduction, and we sharpened abstract and conclusions. Moreover we moved large parts of the model evaluation to the supplementary material, expanded our analysis to JJA 2017 instead of July 2017, added the region Iberian Peninsula in all analyses and shortened the description of previous analyses. In addition, we added a new Section 6 which analyses the ozone regimes in more detail.

You find the changes in the diff version of the manuscript on the following pages:

- page 1 - page 2, l39; revised abstract

- page 3, l72 - page 6, l162 ; revised introduction

- page 9, l237 - page 14 l375 ; shortened evaluation

- page 17, l436 - page 19 l519 ; sharpened analyses of seasonal mean contributions

- page 24, l520 - p 28 ; sharpened analyses of contributions during episodes of large ozone values

- page 29 - page 30 ; new section 6

- page 34, l726 - page 36, l802; revised conclusion

**Specific comments:**

*The introduction discusses at length the difference between perturbation analysis and attribution by tagging. This is an old story and I don't find it particularly interesting. There's nothing wrong with tagging, it just shouldn't be interpreted as a linear response to a perturbation, and we can leave it at that. It would seem more appropriate for the intro to review past relevant studies on attribution of ozone pollution in Europe - this is lacking.*

Answer: We agree that the difference between tagging and perturbation is an "old story", however, from discussions during the review process of previous publications and from discussions on conferences etc. we have the feeling that there is still a lack of understanding of the differences in large parts of the scientific community. Also the second referee asks for results on different mitigation scenarios and why source attribution matters. In that view, we are a bit hesitating to remove the (indeed repeated) discussion. To find a compromise between the comments from the two referees we rephrased the part in the introduction.

You find these changes on page 3, l71 - page 5, l157 of the diff document.

*Although the writing is generally fine (albeit tedious), there are a lot of minor grammatical mistakes and typos that could be corrected by a copy editor*

Answer: For the revision we checked for typos and grammar, but in any case the manuscript will undergo a final check by the journal.

*Page 10, line 12: under- rather than overestimated? There is general ambiguity in referring to frequencies as being under- or over-estimated.*

Answer: This sentence has been moved to the Supplement and rephrased there.

*Page 10, line 19: surface ozone overestimate is attributed to excessive downward mixing but the model also seems to be too high in the free troposphere based on the aircraft comparisons.*

Answer: Yes, indeed, we do have a positive bias of the model results compared to observations. However, we do not see a contradiction here, since an overestimated downward transport will result also in a free tropospheric bias. We revised the overall paragraph:
One main reason for the **positive ground-level** ozone bias is a too strong vertical mixing during the night, **mixing in ozone-rich air from the free troposphere to the boundary layer**. This is is a common problem in many models (Travis and Jacob, 2019). **Moreover, also free-tropospheric ozone is biased high (see disucssuion by e.g. Jöckel et al., 2016).** In consequence, simulated contributions from the stratosphere, from lightning, and from $N_2O$ decomposition to ground level ozone are likely biased high.

*Page 20, line 26: I'm surprised that ozone would be produced from ships by ship NMHCs. My understanding is that the ozone production efficiency from ship NOx emissions in models is very high because the chemistry is strongly NOx-limited, unless some specific model parameterization is used to age the NOx faster but that doesn't seem to be used here. I don't think that ship NMHC emissions are needed – there is plenty of CO and methane around for ozone production in the NOx-limited regime. I may be wrong but a reference would be helpful.*

Answer: As described in the manuscript, the ozone is not produced by ship NMHCs. The ozone is produced by reactions of ship $NO_y$ with NMHCs from evaporation of gas/oil transported (i.e. leaking) **on board** the ships. Indeed, most ozone is formed by $NO_y$ from the ships with $HO_2$. Please keep in mind that we perform source attribution of $NO_x$ **and** NMHCs **concurrently**, implying that we would not see this effect with a pure $NO_x$ attribution method. By revising the colour scheme as asked by referee#2, it should now be much clearer that this production of ozone from NMHCs along the ship lines are only a secondary effect.

**References**

Patrick Jöckel, Holger Tost, Andrea Pozzer, Markus Kunze, Oliver Kirner, Carl Brenninkmeijer, Sabine Brinkop, Duy Cai, Christoph Dyroff, Johannes Eckstein, Franziska Frank, Hella Garny, K. Gottschaldt, Phoebe Graf, Grewe Volker, Astrid Kerkweg, Bastian Kern, Sigrun Matthes, Mariano Mertens, and Andreas Zahn. Earth system chemistry integrated modelling (escimo) with the modular earth submodel system (messy) version 2.51. *Geoscientific Model Development*, 9:1153–1200, 03 2016. doi: 10.5194/gmd-9-1153-2016.

K. R. Travis and D. J. Jacob. Systematic bias in evaluating chemical transport models with maximum daily 8 h average (mda8) surface ozone for air quality applications: a case study with geos-chem v9.02. *Geoscientific Model Development*, 12(8):3641–3648, 2019. doi: 10.5194/gmd-12-3641-2019. URL https://gmd.copernicus.org/articles/12/3641/2019/.